# ReDDiT: Rehashing Noise for Discrete Visual Generation

**Tianren Ma**[1]  **Xiaosong Zhang**[1]  **Boyu Yang**[2]  **Junlan Feng**[2]  **Qixiang Ye**[1]*

[1]University of Chinese Academy of Sciences [2]JIUTIAN Research

matianren18@mails.ucas.ac.cn   qxye@ucas.ac.cn

## ABSTRACT

In the visual generative area, discrete diffusion models are gaining traction for their efficiency and compatibility. However, pioneered attempts still fall behind their continuous counterparts, which we attribute to noise (absorbing state) design and sampling heuristics. In this study, we propose a rehashing noise approach for discrete diffusion transformer (termed **ReDDiT**), with the aim to extend absorbing states and improve expressive capacity of discrete diffusion models. ReDDiT enriches the potential paths that latent variables traverse during training with randomized multi-index corruption. The derived rehash sampler, which reverses the randomized absorbing paths, guarantees high diversity and low discrepancy of the generation process. These reformulations lead to more consistent and competitive generation quality, mitigating the need for heavily tuned randomness. Experiments show that ReDDiT significantly outperforms the baseline model (reducing gFID from 6.18 to **1.61**) and is on par with the continuous counterparts. The code is available at https://github.com/martian422/ReDDiT.

## 1 INTRODUCTION

Diffusion has been a competitive approach for generative workloads (Dhariwal & Nichol, 2021; Rombach et al., 2022b; Li et al., 2024), offering strong bidirectional perception and well-structured mechanisms Zhang et al. (2023) for global control over content. Within the continuous domain, diffusion transformers (DiTs) Peebles & Xie (2023), which progressively refine image latents from Gaussian noise, have achieved impressive and scalable results. Recently, the community shows a growing interest in discrete diffusion models (Hu & Ommer, 2024; Swerdlow et al., 2025), which is based on their practical advantages, *e.g.*, compatibility with language models for the indexable codebook, and efficiency for predicting multiple tokens at each inference. Early endeavors Chang et al. (2022; 2023); Gu et al. (2022) pursue efficiency through integrating visual tokenizers and BERT-style `[mask]` tokens Devlin et al. (2019). Recent studies Bai et al. (2025) improved the generation quality, demonstrating great potential of discrete diffusion.

Despite the progress, the performance of discrete diffusion methods remains lagging behind their continuous counterparts. Representative approaches, *e.g.*, masked visual token models (MVTMs) Chang et al. (2022); Yu et al. (2023), are puzzled by the mask design and confidence-based re-mask sampler (Hur et al., 2024), which restricts model's expressive capacity and makes prediction sensitive to adaptions given extensive training, Fig. 1(upper). Moreover, when paired with large-vocabulary codebooks from high-fidelity modern tokenizers, they encounter challenges such as slower sampling speeds and numerical inaccuracy (Zheng et al., 2024).

To address these limitations, we first propose two hypotheses. First, while discrete methods learn to recover plausible tokens from a monotonous `[mask]` canvas, the used noise design may not be well-suited for discrete visual generation. In continuous diffusion, Gaussian noise is used to progressively degrade the input to learn a smooth distribution shift (Ho et al., 2020; Lu et al., 2022). Discrete masking mimics this paradigm by collapsing all masked tokens to a single absorbing state, which, however, lacks the variability of Gaussian noise, in terms of both vocabulary richness and latent diversity. Consequently, the discrete process offers a far coarser signal, which limits its ability to represent diverse data distributions (Santos et al., 2023; Austin et al., 2021). Moreover, while continuous diffusion models introduce stochasticity at every inference step through noise injection,

---

*Corresponding author.

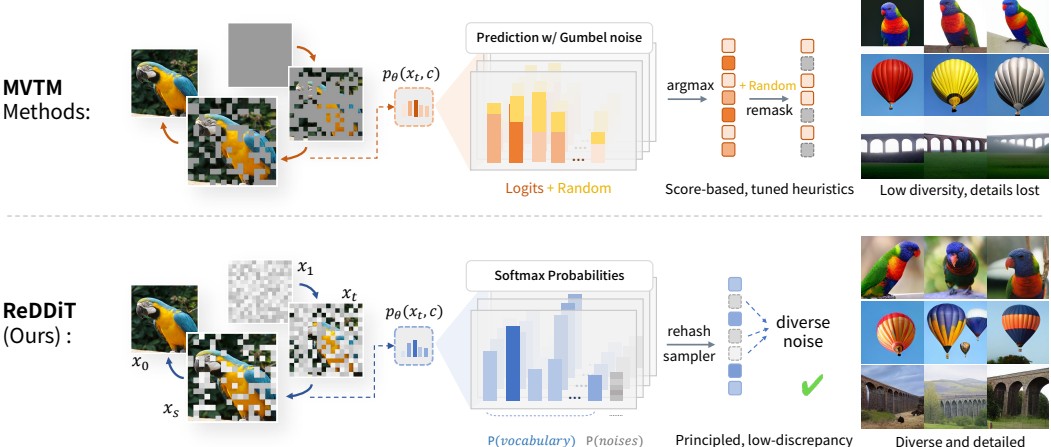

Figure 1: **Comparison the baseline discrete model (MVTM) with ReDDiT.** MVTMs rely on score-based remasking strategies with Gumbel-max to sample from logits, which leads to lower token diversity and suboptimal token selection. In contrast, ReDDiT introduces a systematic, low-discrepancy rehashing mechanism that leverages softmax-based probabilities, enabling diverse, high-quality sampling through a learned distribution. (This figure is best viewed in color)

discrete unmasking is inherently binary: tokens are either masked or deterministically decoded, Fig. 1(upper). This rigid mechanism constrains the flexibility of sample refinement during generation.

Second, the confidence-based re-mask sampler of MVTMs introduces a form of handcrafted randomness, which is implemented through Gumbel-max, to approximate sampling diversity. Unfortunately, this sampler compromises the probabilistic fidelity of generation, and the need to carefully balance token numbers decoded per step (for mitigating accumulation errors) leads to redundant sampling passes. As a result, Gumbel-max has evolved to a heavily tuned time variant trick with unstable performance, particularly when scaled to large-vocabulary codebooks. The above factors, rather than quantization alone, induce the performance gap between discrete and continuous models.

In this study, we propose a discrete diffusion model with an elaborate rehashing noise design, Fig. 1(lower). Our approach, termed **ReDDiT**, addresses the limitations of the uni-mask design by redefining absorbing states towards larger representational capacity, through enriching the potential paths that latent variables can traverse during diffusion. Specifically, we expand the masks to multiple indices along with the codebook and randomize them during data corruption. A rehash sampler is also derived with principled discrete diffusion theories to reverse the diffusion path for generation, guaranteeing high diversity and low discrepancy of the sampling process. We demonstrate that this rehashed noise facilitates learning a superior and regularized expressiveness, while eliminating reliance to hyper-parameterized randomness during sampling.

We further revisit the commonly used discrete diffusion objective and update it with empirical modifications. By adopting an improved ELBO with representation alignment (RepA) Yu et al. (2025) loss, we optimize the training efficiency and substantially improve the generation quality of discrete generative models. Moreover, ReDDiT aligns with recent advances in discrete flow matching Gat et al. (2024); Shaul et al. (2024), enabling token refreshment during sampling without training post-correction models (Lezama et al., 2022).

## 2 METHODOLOGY

For self-containment, we first review the DDM theory in Sec. 2.1. We then reformulate its diffusion dynamics and introduce rehashing noise for ReDDiT in Sec. 2.2. We finally discuss connection and comparison with other discrete diffusion models in Sec. 2.3.

## 2.1 PRELIMINARY: DISCRETE DIFFUSION MODEL

DDM defines a forward process over discrete variables by gradually corrupting the image tokens to absorbing states (masks) through a continuous-time Markov process. Assume that the data consists of tokens from a finite vocabulary $\mathcal{V}$. $x \in \mathcal{V}^L$ is a sequence of tokens (*e.g.*, an image tokenized into indices) with length $L$. We denote the clean data as $x_{t=0}$ ($x_0$ for short), and noise it gradually as $t \to 1$. DDM defines an absorbing token $\mathbf{m} \in \mathcal{V}$, such that once a token is noised to $\mathbf{m}$ it remains unchanged. At the terminal time $t = 1$, $x_t$ fully transits to $\mathbf{m}^L$, which means $x_1^{i=1 \sim L} = \mathbf{m}$.

Let $\alpha_t$ be the noise scheduler (a monotonically decreasing survival function that satisfies $\alpha_0 = 1, \alpha_1 = 0$ ). For $0 \leq s < t \leq 1$, the forward corruption process is governed by a continuous-time transition kernel $q(x_t^i | x_s^i)$ at the $i$-th element, as

$$q(x_t^i | x_s^i) = \begin{cases} 1 - \alpha_{t|s}, & \text{if } x_t^i = \mathbf{m}, x_s^i \neq \mathbf{m} \\ \alpha_{t|s}, & \text{if } x_t^i = x_s^i, x_s^i \neq \mathbf{m} \\ 1, & \text{if } x_t^i = x_s^i, x_s^i = \mathbf{m} \\ 0, & \text{otherwise} \end{cases}, \quad \alpha_{t|s} = \frac{\alpha_t}{\alpha_s}. \tag{1}$$

Denoting $q$ as the transition kernel and $\text{Cat}(\cdot; \pi)$ the categorical distribution determined by probability $\pi$, the corrupted data distribution at time $t$ is written as

$$x_t \sim q(x_t | x_0), q(x_t | x_0) = \text{Cat}(x_t; \alpha_t x_0 + (1 - \alpha_t) \mathbf{m}^L). \tag{2}$$

The generative model learns the reverse process $p_\theta(x_s | x_t)$, which denoises sample $x_t$ at arbitrary time $t \in (0, 1]$ to a less noised state $x_s$ at time $s < t$. Denoting $\delta(x_t^i, \mathbf{m})$ as the indicator function that only computes on masked tokens, and $\alpha_t' = \frac{d\alpha_t}{dt}$, the learning objective is derived as

$$\mathcal{L}_{\text{DDM}} = -\mathbb{E}_{x_0, x_t} \int_{t=0}^{t=1} \left[ \frac{\alpha_t'}{1 - \alpha_t} \sum_{i=1}^{L} \delta(x_t^i, \mathbf{m}) \log p_\theta(x_0^i | x_t) \right] dt . \tag{3}$$

For a linear scheduler, Eq. 3 is simplified via variable substitution Sahoo et al. (2024) to an equivalent form, as

$$\mathcal{L}_{\text{DDM-linear}} = -\mathbb{E}_{t, x_0, x_t} \left[ \frac{1}{t} \sum_{i=1}^{L} \delta(x_t^i, \mathbf{m}) \log p_\theta(x_0^i | x_t) \right] . \tag{4}$$

For conditional generation, class information $c$ (*e.g.*, labels or text prompts) is introduced to the denoising model as additional input. Following classifier-free guidance Ho & Salimans (2022), the model is trained with a random drop of labels, and the prediction is interpolated at inference, as

$$\hat{p}_\theta(x_t, c) = p_\theta(x_t, \varnothing) + w \cdot (p_\theta(x_t, c) - p_\theta(x_t, \varnothing)), \tag{5}$$

where $\varnothing$ is the dropped label and $w \geq 0$ controls the guidance strength.

## 2.2 DISCRETE DIFFUSION WITH REHASHING NOISE

The ordinal structure inherent in discrete data provides a valuable inductive bias for designing transition kernels in diffusion dynamics. Prior studies Austin et al. (2021); Campbell et al. (2022) show that assigning higher transition probabilities to neighboring pixel values—forming a *discrete Gaussian-like noise*—outperforms the single absorbing state approach on pixel-level datasets like CIFAR-10. However, when using visual tokenizers, the structure of discretized latents is learned rather than pre-defined, making such ordinal assumptions inapplicable. This insight motivates us to extend conventional mask tokens to a set of indices, and reverse the diffusion path with noise rehashing. This design allows the model to optimize its embedding space during training, enhancing its ability to model flexible and data-driven noise structures. We visualize the learned distributions in Fig. 2 (right).

**Reformulation.** Given $d$ categories, let $\mathbf{e}_i \in \mathbb{R}^d$ be its one-hot vector where the $i$-th value is 1. We denote $\mathcal{E} = \{\mathbf{e}_i \in \mathbb{R}^d \mid i = 1, \ldots, d\}$ as the basis of a categorical distribution (known as $d$-simplex), and a basis for absorbing states with capacity $m$: $\mathcal{M} = \{\mathbf{m}_j \in \mathbb{R}^m \mid j = 1, \ldots, m\}$. With subscript $i, j = 0$ for pure visual or mask space, the sum of $\mathcal{E}$ and $\mathcal{M}$ can be denoted as

$$\mathcal{V}_{(d,m)} \triangleq \left\{ \mathbf{v}_{(i,j)} \in \mathbb{R}^{d+m} \,\middle|\, \mathbf{v}_{(i,j)} = \begin{cases} \mathbf{e}_i \oplus \mathbf{0}_m, & \text{for } i = 1, \ldots, d, \ j = 0 \\ \mathbf{0}_d \oplus \mathbf{m}_j, & \text{for } j = 1, \ldots, m, \ i = 0 \end{cases} \right\}. \tag{6}$$

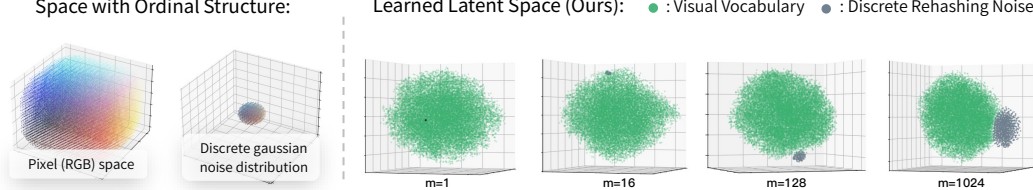

Figure 2: **Visualization of pixel and latent spaces.** $m$ denotes the number of enriched noise indices. Note that the 3D t-SNE plot (right) is used solely for clustering with no isotropic features.

We further denote the subspace $\mathcal{E}_d$, $\mathcal{M}_m \in \mathcal{V}_{(d,m)}$ which contain *either* valid or mask tokens, as

$$\mathcal{E}_d = \left\{ \mathbf{v}_{(i,0)} \in \mathcal{V}_{(d,m)} \,\middle|\, i = 1, \ldots, d \right\}, \quad \mathcal{M}_m = \left\{ \mathbf{v}_{(0,j)} \in \mathcal{V}_{(d,m)} \,\middle|\, j = 1, \ldots, m \right\}. \quad (7)$$

To exploit visits across all the possible paths, we rewrite the transition kernel defined by Eq. 1 as

$$q(x_t^i | x_s^i) = \begin{cases} 1 - \alpha_{t|s}, & \text{if } x_t^i \in \mathcal{M}_m, \ x_s^i \notin \mathcal{M}_m \\ \alpha_{t|s}, & \text{if } x_t^i = x_s^i, \ x_s^i \notin \mathcal{M}_m \\ 1/m, & \text{if } x_t^i \in \mathcal{M}_m, \ x_s^i \in \mathcal{M}_m \\ 0, & \text{otherwise.} \end{cases} \quad (8)$$

With above definitions, we reformulate the diffusion process of $x$ as a **transition from $\mathcal{E}_d$ to $\mathcal{M}_m$**. We train the model by feeding it with corrupted data, of which the distribution is inferred as $x_t \sim \text{Cat}(x_t; \alpha_t x_0 + (1 - \alpha_t)\text{U}(\mathcal{M}_m^L))$, where $\text{U}(\mathcal{M}_m^L)$ is the uniform distribution upon $\mathcal{M}_m^L$.

**Rehash Sampling.** To generate a sequence of length $L$, the reverse process starts with $x_1 \sim \text{U}(\mathcal{M}_m^L)$. The subsequent latents $x_t$ are generated by discretizing the reverse timeline $T$ to $K$ steps. We denote this schedule as $T^{1:K+1}$ such that $T^1 = 1$ and $T^{K+1} = \varepsilon$, with $\varepsilon$ being an arbitrarily small positive constant. Let $\delta^i$ indicate the $i$-th token's value, the reverse process is deduced as

$$q_{s|t}^i = q(x_s^i | x_t) = \begin{cases} 1, & \text{if } x_s^i = x_t^i, \ x_t^i \notin \mathcal{M}_m \\ \frac{1 - \alpha_s}{m(1 - \alpha_t)}, & \text{if } x_s^i \in \mathcal{M}_m, \ x_t^i \in \mathcal{M}_m \\ \frac{\alpha_s - \alpha_t}{1 - \alpha_t} \delta^i p_\theta(x_t), & \text{if } x_s^i \notin \mathcal{M}_m, \ x_t^i \in \mathcal{M}_m \\ 0, & \text{otherwise.} \end{cases} \quad (9)$$

Comparing with MVTM sampler in Alg. 1, our rehash sampler is shown in Alg. 2. Our algorithm shares the similar idea as MDLM Sahoo et al. (2024), but applies `torch.multinomial` (Multnm. in step 10) for low-discrepancy[1] categorical sampling.

---

**Algorithm 1** MVTM Sampling

1: **Inputs:** label $c$, scheduler $\alpha_t$, length $L$,
2: **Settings:** number of steps $K$, $G(t)$, $\mathcal{G}$
3: Initialize: $x_1 \leftarrow \mathcal{M}_1^L$, $t \leftarrow 1$.
4: **for** $k = 1$ to $K$ **do**
5:      $t \leftarrow \frac{K-k+1}{K}$, $s \leftarrow \frac{K-k}{K}$
6:      $\text{logits} \leftarrow f_\theta(x_t, c)$
7:      $p_{\text{score}} \leftarrow \text{logits} + G(t) \cdot \mathcal{G}$
8:      $x_{\text{pred}} \leftarrow \text{argmax}(p_{\text{score}})$    ▷ Predict-all
9:      $x_s \leftarrow \text{where}(x_t = [m], x_{\text{pred}}, x_t)$
10:    $p_{\text{conf}} \leftarrow p_{\text{score}} + G(t) \cdot \mathcal{G}$
11:    $m_{\text{re}} \leftarrow \text{argsort}(p_{\text{conf}})[1 : L \cdot (1 - \alpha_s)]$
12:    $x_s \leftarrow \text{where}(m_{\text{re}}, [m], x_s)$   ▷ Re-mask
13: **end for**
14: **Return:** fully unmasked sequence $x_0$

**Algorithm 2** Rehash Sampling (Ours)

1: **Inputs:** label $c$, scheduler $\alpha_t$, length $L$.
2: **Settings:** number of steps $K$.
3: Initialize: $x_1 \sim \text{U}(\mathcal{M}_m^L)$, $t \leftarrow 1$, $T^{1:K}$.
4: **for** $k = 1$ to $K$ **do**
5:      $t \leftarrow T^k$, $s \leftarrow T^{k+1}$
      *# the rehash operation:*
6:      $x_t \leftarrow \text{where}(x_t \in \mathcal{M}_m, \text{U}(\mathcal{M}_m^L), x_t)$
7:      $\text{logits} \leftarrow f_\theta(x_t, c)$
8:      $p \leftarrow \text{Softmax}(\text{logits})$
9:      $q_{s|t} \leftarrow \frac{\alpha_s - \alpha_t}{1 - \alpha_t} \cdot p + \delta_{m[0]} \cdot \frac{1 - \alpha_s}{1 - \alpha_t}$
10:    $x_{\text{pred}} \leftarrow \text{Multnm.}(q_{s|t})$    ▷ w/ masks
11:    $x_s \leftarrow \text{where}(x_t \in \mathcal{M}_m, x_{\text{pred}}, x_t)$
12: **end for**
13: **Return:** fully unmasked sequence $x_0$

---

[1] MDLM uses gumbel-max for sampling, which may incur inaccuracy (Zheng et al., 2024). Besides, we deliberately merge the probabilities at step 9 to keep an overall noise sampling probability, as small values might be truncated, which also worsens sampling accuracy.

The random nature of absorbing states inspires a rehash operation: we shuffle these tokens at the beginning of each step by $x_t \leftarrow \text{where}(x_t \in \mathcal{M}_m, \text{U}(\mathcal{M}_m^L), x_t)$. Proof to Eq.9 is included in Appendix. B.

## 2.3 DISCUSSION

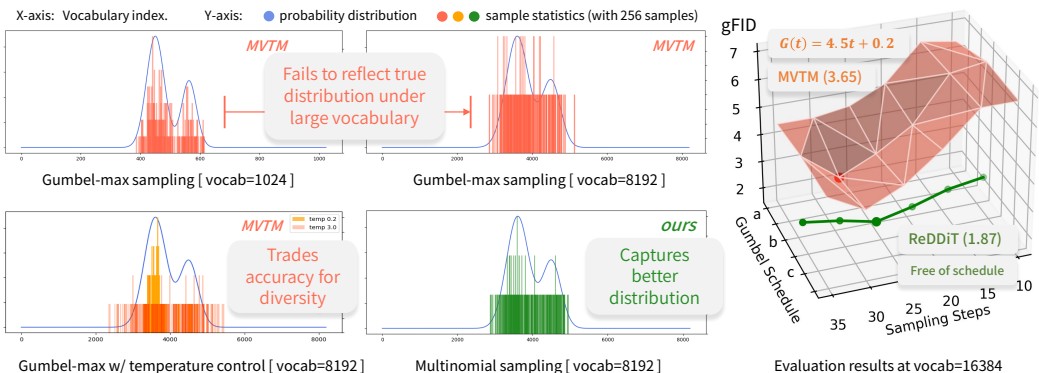

Figure 3: **Sampler comparison.** *Left*: Gumbel-max is theoretically equivalent to our method, yet it struggles to reflect the true distribution under limited sample passes. The multinomial approach captures the distribution more accurately. *Right*: our model achieves lower gFID across different sampling steps without tuning Gumbel-max, indicating more efficient and faithful sampling. *a, b, c* refer to three uniformly sampled $G(t)$ set for MVTM sampling. See supplementary for experimental codes. (This figure is best viewed in color)

**Comparison with MVTM.** Masked visual token models (MVTMs) borrow the objective

$$\mathcal{L}_{\text{MVTM}} = -\mathbb{E}_{t,\, x_0,\, x_t} \sum_{i=1}^{L} \delta(x_t^i, \mathbf{m}) \log p_\theta(x_0^i \mid x_t),$$ (10)

from masked language models Devlin et al. (2019) and predict on masked tokens with a maximum likelihood. Besides the reformulated corruption (Eq. 8) and reverse process (Eq. 9), ReDDiT differs in the following aspects: (*i*) the training objective (Eq. 4), which is derived from DDM, providing better theoretical and empirical results. (*ii*) it can easily sample with a arbitrarily discretized timeline, while MVTM couples training and inference, restricting its sampling flexibility; (*iii*) the rehash sampler (Alg. 2) includes absorbing states in categorical sampling with lower discrepancy, different from MVTM's predict-remask sampler with time variant intensity $G(t)$ over Gumbel noise $\mathcal{G}$ (Alg. 1) [2]. Gumbel-max suffers from numerical inaccuracy (Zheng et al., 2024) and we noitice that it becomes worse on large vocabulary (Fig. 1, 3 with our reproduced results), which limits MVTM's potential.

**Relationship to DFM.** Discrete flow matching (DFM) Gat et al. (2024) introduces a transition process based on masked tokens. Its training objective was initially designed as the masked token loss ( 10), and evolved to a time-weighted cross-entropy loss (Shaul et al., 2024) for generalized diffusion paths, which is similar to ours. The similarity enables a direct comparison between the DFM sampler and our rehash sampler using the same trained model weights. We notice that it generally requires more steps to reach optimal results, as the DFM sampler offers a refinement mechanism via token-wise updates. Since the gradual decoding method is shared, we can integrate certain DFM steps into our sampling procedure for refinement. This leads to $\sim 0.1$ gFID improvement on ImageNet-1K. Refer to Appendix D for details.

---

[2]The logits corresponding to previously restored tokens' indices are manually set to infinity for both methods, so that they will not be noised again in the following steps. This leads to an implementation of any-order auto-regressive model (Ou et al., 2024) if DDM's decoded tokens per step is limited to 1.

## 3 EXPERIMENT

### 3.1 IMPLEMENTATION

**Datasets.** The experiments are conducted on ImageNet-1K Deng et al. (2009), which consists of 1000 categories, 1281167 images and are cropped to resolution $256 \times 256$ for training. The generation quality is evaluated using Fréchet Inception Distance (FID) Heusel et al. (2017) and the Inception Score (IS) Salimans et al. (2017). FID measures the distance between the distributions of generated and real images in the feature space of a pre-trained Inception network, while IS evaluates both the confidence and diversity of generated images by analyzing predicted label distribution. We compute generation FID (gFID↓)[3] and IS↑ on 50k generated samples.

**Pre-processing.** Following the setting in LlamaGen Sun et al. (2024), we apply the ten-crop augmentation on images, and use pre-trained tokenizers to convert them to discrete tokens. We pick IBQ-f16 Shi et al. (2025) tokenizer as default for its scalable and promising performance in generation tasks, which uses a $16 \times 16$ downsampling ratio and converts a $256 \times 256$ image into 256 discrete tokens. The tokenizer has a codebook with 16384 entries. The LlamaGen-f16 (used in Tab. 2) and LlamaGen-f8 tokenizer Sun et al. (2024) (used in Tab. 1) are also used for comparison with recent discrete generation methods. All tokenizers are used out-of-the-box without modification.

**Representation Alignment.** Recent study Yu et al. (2025) has shown that the alignment of intermediate representations between diffusion transformers and vision encoders accelerates training convergence of diffusion models. Accordingly, the alignment is designed as a regularization term with $\lambda = 0.5$. We extract diffusion transformer's $8$-th layer intermediate feature $\mathbf{h}^{[n]}(x_t)$ and align it with the original image's dinov2-b Oquab et al. (2023) encoded features $f_{\text{enc}}(x_0^{\text{ori}})$. The intermediate features are projected by a small trainable MLP $h_\varphi$. The $\text{sim}(\cdot, \cdot)$ computes the mean of element-wise cosine similarity between embeddings, as

$$\mathcal{L}_{\text{total}} = \mathcal{L}_{\text{DDM-linear}} + \lambda \mathcal{L}_{\text{RepA}}, \quad \mathcal{L}_{\text{RepA}} = -\mathbb{E}_{x,\,t}\big[\,\text{sim}(f_{\text{enc}}(x_0^{\text{ori}}),\,h_\varphi(\mathbf{h}^{[n]}(x_t)))\,\big]\,. \quad (11)$$

This alignment was proposed for continuous diffusion models, and we firstly validate that it's also suitable for training discrete models. However, from our observation, as a training acceleration technique, RepA **does not** provide relative performance gain if training sufficiently for discrete latents. We only use RepA to improve training efficiency and probe the inner dynamics through training as in Fig. 4. See Appendix F for a detailed discussion.

**Training and Evaluation.** The proposed model is based on DiT Peebles & Xie (2023) architecture, with reference to its discrete prediction version Sahoo et al. (2024). 2D-RoPE Su et al. (2024) and min-SNR Zhang & Sennrich (2019) are applied for training efficiency. The model is optimized using the AdamW optimizer with a cosine decay. Class-conditional training is enabled using class embeddings and a drop-rate of 0.1 for generation with CFG. Details are provided in Appendix E.

### 3.2 PERFORMANCE AND COMPARISON

We compare the proposed ReDDiT model with other generative models on the ImageNet-1K $256 \times 256$ in Tab. 1. The IBQ tokenizer is used for the default L and XL models. We also utilize LlamaGen-f8 with 128 noise capacity to evaluate its high-resolution potentials (noted as ReDDiT-XL$_{\text{f8}}$). We use a linear increasing guidance following the common practice of Gao et al. (2023).

**Generation Quality.** As shown in Tab. 1, ReDDiT achieves the best performance among the compared discrete models. It outperforms the baseline (MaskGIT Chang et al. (2022)) with significant margins (gFID: 2.13 vs 6.18 and IS: 294.7 vs. 182.1). It also outperforms the recent DDM method Hu & Ommer (2024) and TiTok-S-128 Yu et al. (2024), which is extensively fine-tuned on quantized latents. Compared with continuous diffusion models, ReDDiT exhibits on-par efficiency and performance, showing great potential for discrete generation.

---

[3]The gFID is used as the quality metric for generative models' performance, while rFID refers to the reconstruction quality of a visual tokenizer.

Table 1: **Performance comparison on class-conditional ImageNet 256×256.** Look-up free quantizers are beyond the scope of this paper. *ft.*(in gray) indicates that the decoder is fine-tuned for quantized latents. Wall-clock inference time relative to ReDDiT-XL is reported.

| Type | Model | Tokenizer | | Generator | | | | |
|------|-------|-----------|--|-----------|--|--|--|--|
| | | #tokens | codebook | gFID↓ | IS↑ | #Params | #Steps | Time |
| Diff. | LDM-4 Rombach et al. (2022a) | 4096×3 | - | 3.60 | 247.7 | 400M | 250 | – |
| | DiT-XL/2 Peebles & Xie (2023) | 1024×4 | - | 2.27 | 278.2 | 675M | 250 | 18 |
| | MDTv2 Gao et al. (2023) | 1024×14 | - | 1.58 | 314.7 | 676M | 256 | 18 |
| | SiT-XL Ma et al. (2024) | 1024×4 | - | 2.42 | 238.5 | 675M | 30 | 2 |
| | SiT-XL w/ Solver Wang et al. (2025) | 1024×4 | - | 2.24 | 244.1 | 730M | 15 | 1.2 |
| AR | Taming-VQGAN Esser et al. (2021) | 256 | 1024 | 15.78 | 74.3 | 1.4B | 256 | 8 |
| | RQ-Transformer Huang et al. (2023) | 256 | 16384 | 7.55 | 134.0 | 3.8B | 64 | 8.5 |
| | ViT-VQGAN Yu et al. (2022) | 1024 | 8192 | 4.17 | 175.1 | 1.7B | 1024 | >10 |
| | LlamaGen-3B Sun et al. (2024) | 576 | 16384 | 2.18 | 263.3 | 3.1B | 576 | 20 |
| | RandAR-XXL Pang et al. (2024) | 512 | 16384 | 2.15 | 322.0 | 1.4B | 88 | 4 |
| | VAR-$d$30 Tian et al. (2024) | 680 | 4096 | 1.97 | 334.7 | 2.0B | 10 | 0.5 |
| MVTM | MaskGIT Chang et al. (2022) | 256 | 1024 | 6.18 | 182.1 | 227M | 8 | 0.2 |
| | MaskGIL-XXL Xin et al. (2025) | 256 | 16384 | 3.71 | 303.4 | 1.4B | 8 | 0.8 |
| | TiTok-S-128$_{ft.}$ Yu et al. (2024) | 128 | 4096 | 1.97 | 281.8 | 287M | 64 | 1.6 |
| DDM | ITM Hu & Ommer (2024) | 1024 | 16384 | 5.30 | 183.0 | 546M | 100 | 3 |
| | ReDDiT-L (ours) | 256 | 16384 | 2.13 | 294.7 | 346M | 20 | 0.5 |
| | ReDDiT-XL (ours) | 256 | 16384 | 1.74 | 313.6 | 675M | 32 | 1 |
| | ReDDiT-XL$_{f8}$ (ours) | 1024 | 16384 | **1.61** | 318.5 | 675M | 64 | 2 |

Table 2: **Comparison of models with the same tokenizer.** Reconstruction FID (rFID) indicates the tokenizer's reconstruction quality from its quantized codes. Dim denotes codebook dimension.

| Model | VQ Tokenizer Info. | | | Generator | |
|-------|--------------------|--|--|-----------|--|
| | Identity | rFID | dim | #Params | gFID↓ |
| LlamaGen-L$_{AR}$ Sun et al. (2024) | | | | 343M | 3.80 |
| RandAR-L$_{AR}$ Pang et al. (2024) | LlamaGen-f16 Sun et al. (2024) | 2.19 | 8 | 343M | 2.55 |
| Ours$_{DDM(ReDDiT-L)}$ | | | | 346M | 2.41 |
| IBQ-B$_{AR}$ Shi et al. (2025) | IBQ-tokenizer Shi et al. (2025) | 1.37 | 256 | 343M | 2.88 |
| Ours$_{DDM(ReDDiT-L)}$ | | | | 346M | 2.13 |

**Efficiency.** ReDDiT is born with the high-efficiency advantage of discrete diffusion models, comparing with AR models. As shown in Tab. 1, the inference time of ReDDiT is slightly longer than MaskGIT, while the performance is overwhelming. Without acceleration techniques, ReDDiT achieves a competitive performance which AR and traditional diffusion models use more than 250 steps to achieve. Notably, when armed with recent efforts that tailored KV-Cache Liu et al. (2025) for discrete diffusion models, ReDDiT's inference can be further boosted (not included in the main paper for fair comparison). See Appendix G for details.

Besides the major comparison, we also conduct an experiment that utilizes the identical tokenizer in previous AR models and validate our method's effectiveness. As can be seen in Tab. 2, ReDDiT outperforms AR methods in generation tasks across different tokenizers.

## 3.3 DETERMINING NOISE CAPACITY

The reformulated discrete diffusion dynamics defines transitioning from $\mathcal{E}_d$ to $\mathcal{M}_m$. Under this setting, it is necessary to empirically determine the optimal value of $m$ for a fixed tokenizer with vocabulary size $d$, as the latent representations learned by VAEs are variant. We keep the training setup fixed and conduct experiments *w.r.t.* the noise capacity $m$. We also visualize $\mathcal{L}_{RepA}$, which captures the degree of representation alignment within the transformer.

The alignment loss visualization shows that increasing the number of absorbing states introduces greater randomness, initially making predictions more difficult due to confusion with valid tokens. However, this gap narrows as training progresses, and the model converges to a similar alignment

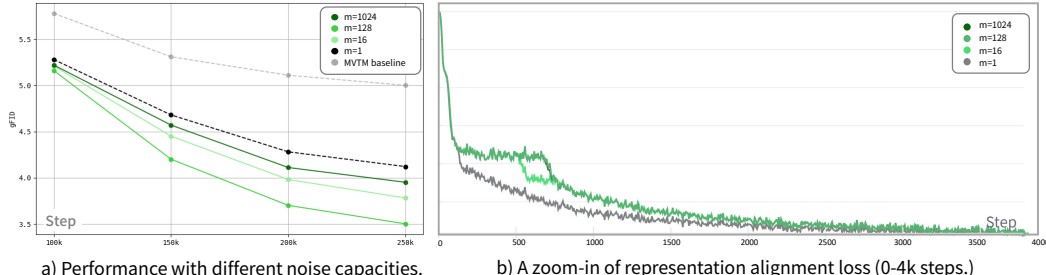

a) Performance with different noise capacities.      b) A zoom-in of representation alignment loss (0-4k steps.)

Figure 4: **Comparison of noise capacities.** We re-implemented training with the same training recipe. The generation quality and representation alignment trends are visualized.

lower bound, suggesting effective representation learning across different configurations. As shown in Fig. 4 (left), generation quality improves with increasing noise capacity initially. The LlamaGen-f16 tokenizer achieves peak performance at $m = 128$, while the IBQ tokenizer performs best at $m = 1024$. We attribute this to the codebook design: the lower dimensional LlamaGen-f16 codebook produces more compact latents, which also determines its smaller noise endurance.

## 3.4 ABLATION STUDY

Unless specified, all the models are trained on ImageNet $256 \times 256$ under the default settings for 100k iterations. We use a constant guidance scale of 2.0 and 20 steps for generation, and report gFID ↓ computed on 50K samples. Precision (Prec.↑) and Recall (Rec.↑) are also reported.

**Sampling Timeline.** Recovering complete information from noise remains critical to diffusion-based models (Lu et al., 2022; Wu et al., 2024). Recent work shows MVTM's non-linear scheduler for training is less critical when using high-capacity tokenizers. Evidence of time-invariance in DDMs (Sahoo et al., 2024; Shi et al., 2024) further supports decoupling training from sampling. In our experiments, a linear scheduler with constant signal-to-noise ratio decay, yields optimal training dynamics. Among the timeline discretization tested, Fig. 5, the *cosine* schedule is employed for our ReDDiT model for best performance in Tab. 3.

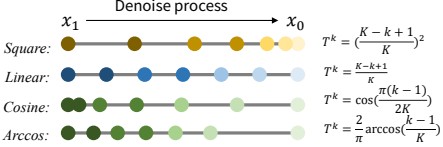

Figure 5: Illustration of discretized timeline with $K = 7$. The slow-to-fast sampling works better than linear schedules.

Table 3: **Ablated Design Choices.** ReDDiT-L is trained for 100k iters. Final setting denoted in gray.

<table>
<tr><td colspan="5" align="center">(a) General model design</td><td colspan="3" align="center">(b) Sampling timeline</td></tr>
<tr><td>Train Config</td><td>Sample Config</td><td>gFID</td><td>Prec.</td><td>Rec.</td><td>Steps</td><td>Timeline</td><td>gFID</td></tr>
<tr><td>MVTM + RepA loss</td><td>MVTM sampler</td><td>*6.83*</td><td>0.75</td><td>0.39</td><td>20</td><td>*linear*</td><td>7.18</td></tr>
<tr><td>Switch to objective (11)</td><td>MVTM sampler</td><td>6.23</td><td>0.77</td><td>0.41</td><td>32</td><td>*linear*</td><td>6.43</td></tr>
<tr><td>same as above</td><td>Rehash sampler</td><td>*5.75*</td><td>0.78</td><td>0.45</td><td>20</td><td>*arccos*</td><td>5.04</td></tr>
<tr><td>+ 2D-RoPE + min-SNR</td><td>Rehash sampler</td><td>5.51</td><td>0.79</td><td>0.45</td><td>20</td><td>*square*</td><td>7.39</td></tr>
<tr><td>same as above</td><td>+ DFM refine</td><td>5.40</td><td>0.81</td><td>0.52</td><td>20</td><td>*cosine*</td><td>4.91</td></tr>
</table>

**General Design.** We ablate the general choices of ReDDiT, which starts with a re-trained MVTM baseline methods (with LlamaGen-f16 and RepA for faster convergence as default) in Tab. 3. The applied techniques like 2D-RoPE are also ablated with re-training. As shown, through the revised objective and our proposed sampler, ReDDiT alone improves FID by $\sim 1.0$ compared to the baseline model. When combined with modern modification on transformers, it can further improve the performance, showing its complementaryness with main-stream efforts.

**Impact of Rehash Operation.** To validate the rehash operation for encouraging path diversity by resampling noise tokens, we compare noise capacities $m = 1$ and $m = 128$. As shown in Table 4, increasing capacity with a fixed absorbing state actually degrades performance compared to the baseline. While enabling random initialization improves gFID, the full rehash mechanism is essential to unlock the model's capacity, confirming that active resampling is required to prevent overly deterministic sampling.

Table 4: Ablation on Rehashing.

| Setting | gFID |
|---|---|
| $m = 1$ (Baseline) | 4.13 |
| $m = 128$ (Fixed State) | 4.25 |
| $m = 128$ (No Rehash) | 4.07 |
| $m = 128$ (Full Rehash) | **3.92** |

### 3.5 QUALITATIVE RESULT

**Class-conditional Generation.** Figure 6 presents representative class-conditional samples generated by the proposed ReDDiT model. The outputs across diverse image classes consistently exhibit high fidelity and diversity. Please refer to Appendix H for more samples.

**Image Editing.** We further demonstrate ReDDiT's editing capability in Figure 6, highlighting its bi-directional perceptual competence. Following MaskGIT, we replace a region of the input image with noise tokens and employ the same generation pipeline to inpaint the missing content, conditioned on a class label $c$.

## 4 RELATED WORK

**Diffusion Models.** Diffusion models Ho et al. (2020); Song et al. (2020) have emerged as a powerful class of generative methods that learn data distributions by reversing a gradual noising process over time. These models are primarily designed for continuous domains such as images Dhariwal & Nichol (2021); Gao et al. (2023); Peebles & Xie (2023), defining a forward process that transforms data $x_0$ into noise $x_1$: $x_t \sim \mathcal{N}(\sqrt{\alpha_t}x_0; (1 - \alpha_t)\mathbf{I})$ where $\alpha_t$ controls the noise schedule. The generative (reverse) process learns a denoising model $p_\theta(x_s \mid x_t)$, often parameterized via a neural network $\theta$ to predict either noise or clean data.

**Discrete Diffusion Models.** Discrete diffusion has been previously governed by masked visual token models (MVTMs) Chang et al. (2022; 2023); Gu et al. (2022); Yu et al. (2023; 2024); Hur et al. (2024). This model leverages a BERT-style `[mask]` token to corrupt the tokenized image sequence and trained the network with a simple cross-entropy loss on masked tokens, resulting in a score-based prediction. It generates tokens in a non-autoregressive fashion, by remasking the tokens with least scores at each inference as depicted in Alg. 1.

Recent studies unlocked the principled discrete diffusion model (DDM) Sahoo et al. (2024); Shi et al. (2024) and discrete flow-matching (DFM) Gat et al. (2024); Shaul et al. (2024), which adapt the Markov chain theory, enabling generation over text Ou et al. (2024); Nie et al. (2025), moleculesShaul et al. (2024), and other discrete representations Austin et al. (2021); Nisonoff et al. (2024). Unlike MVTMs, the principled DDM and DFM mostly derive a time-weighted cross-entropy loss to supervise the training procedure and apply a gradual unmasking method based on probabilities.

## 5 CONCLUSION

We proposed ReDDiT, a discrete visual generative model built upon a discrete diffusion architecture with novel noise designs and efficient sampling strategies. Our key contribution lies in the integration of rehashing noise with samplers, which together ensure both diversity and low discrepancy throughout the generative process. By introducing rehashing noise, ReDDiT enriches the potential paths that latent variables can traverse during training, regularize training dynamics and enhances model's representational capacity. Extensive experiments demonstrate that discrete generative models can achieve performance on par with their continuous counterparts while offering top-tier efficiency. This study paves a promising way for discrete generative modeling and offers fresh insights toward unifying visual and language generation—a path we leave for future exploration.

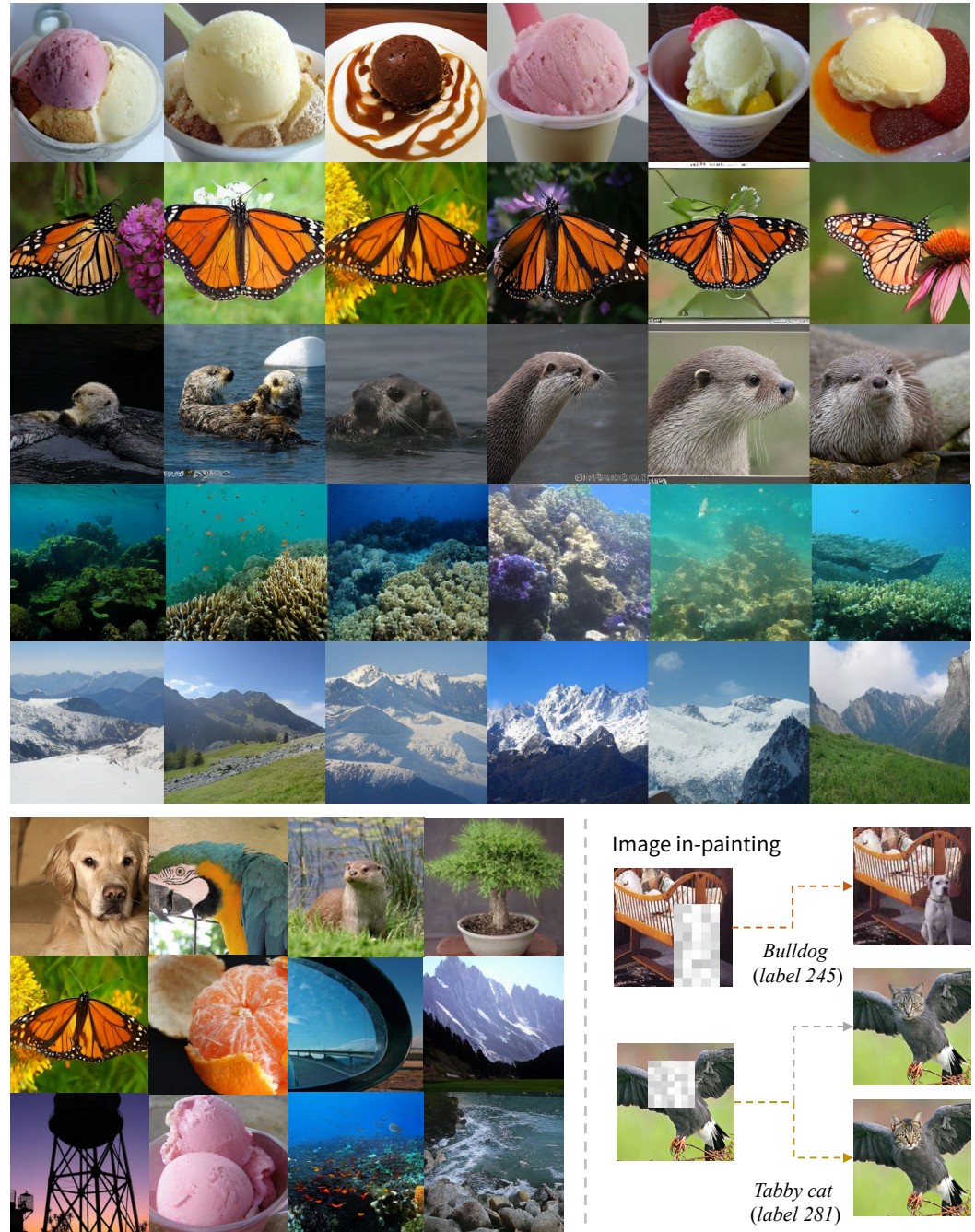

Figure 6: Class-conditional generation and in-painting samples of ReDDiT on ImageNet $256 \times 256$.

**Reproducibility** We have provided key algorithms in the main text. Further implementation details are available in the source code.

## 6 ACKNOWLEDGMENTS

The authors would like to thank JIUTIAN Research for their computational resources. This work was supported by National Natural Science Foundation of China under Grant 62225208, 62521007 and CAS Project for Young Scientists in Basic Research under Grant No.YSBR-117.

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

## A  ETHICAL STATEMENTS

In the process of drafting this paper, Large Language Models (LLMs) were solely utilized for **grammatical checking**. No LLM was involved in core academic work such as conceptualization, literature review, data analysis, or argument construction of this study.

As the human authors of this paper, we bear full and sole responsibility for the paper's content, including the accuracy of research data, validity of academic arguments, integrity of research methods, and compliance with academic ethics.

## B   DISCRETE DIFFUSION WITH REHASHING NOISE

**Complete Definition and Deduction.**   We provide a full theoretical discussion on the corrupted distribution and reverse process defined in the main paper.

Given $d$ categories, let $\mathbf{e}_i \in \mathbb{R}^d$ be its one-hot vector where the $i$-th value is 1. We denote $\mathcal{E} = \{\mathbf{e}_i \in \mathbb{R}^d \mid i = 1, \ldots, d\}$ as the basis of a categorical distribution, and a basis for absorbing states with capacity $m$: $\mathcal{M} = \{\mathbf{m}_j \in \mathbb{R}^m \mid j = 1, \ldots, m\}$. The sum of $\mathcal{E}$ and $\mathcal{M}$ can be denoted as

$$\mathcal{V}_{(d,m)} \triangleq \left\{ \mathbf{v}_{(i,j)} \in \mathbb{R}^{d+m} \; \middle| \; \mathbf{v}_{(i,j)} = \begin{cases} \mathbf{e}_i \oplus \mathbf{0}_m, & \text{for } i = 1, \ldots, d, \; j = 0 \\ \mathbf{0}_d \oplus \mathbf{m}_j, & \text{for } j = 1, \ldots, m, \; i = 0 \end{cases} \right\}. \tag{12}$$

We further denote the subspace $\mathcal{E}_d, \mathcal{M}_m \in \mathcal{V}_{(d,m)}$ which contain either valid or mask tokens, as

$$\mathcal{E}_d = \left\{ \mathbf{v}_{(i,0)} \in \mathcal{V}_{(d,m)} \, \middle| \, i = 1, \ldots, d \right\}, \; \mathcal{M}_m = \left\{ \mathbf{v}_{(0,j)} \in \mathcal{V}_{(d,m)} \, \middle| \, j = 1, \ldots, m \right\}. \tag{13}$$

To exploit visits across all the possible paths, for $0 \le s < t \le 1$, we write the transition kernel as[4]

$$q(x_t^i \mid x_s^i) = \begin{cases} 1 - \alpha_{t|s}^{\leftarrow}, & \text{if } x_t^i \in \mathcal{M}_m, \; x_s^i \notin \mathcal{M}_m, \\ \alpha_{t|s}^{\leftarrow}, & \text{if } x_t^i = x_s^i, \; x_s^i \notin \mathcal{M}_m, \\ 1/m, & \text{if } x_t^i \in \mathcal{M}_m, \; x_s^i \in \mathcal{M}_m, \\ 0, & \text{otherwise.} \end{cases} \tag{14}$$

**Proof of the Corrupted Distribution.**   The presentation in the main paper simplifies the theory without specifying the transition matrix $Q_t$ due to page limitation. We make a detailed version with important yet basic matrix calculation in this section.

Let $\mathbf{I}_{(d,m)}$, $\mathbf{M}_{(d,m)}$ and $\boldsymbol{\pi}_{(d,m)}$ be matrices in $\mathbb{R}^{(d+m)\times(d+m)}$, defined as

$$\mathbf{I}_{(d,m)} = \begin{bmatrix} I_d & 0 \\ 0 & 0 \end{bmatrix}, \quad \mathbf{M}_{(d,m)} = \begin{bmatrix} 0 & \frac{1}{m}\mathbf{1}_{d\times m} \\ 0 & 0 \end{bmatrix}, \quad \boldsymbol{\pi}_{(d,m)} = \begin{bmatrix} 0 & 0 \\ 0 & \frac{1}{m}\mathbf{1}_m\mathbf{1}_m^\top \end{bmatrix} \tag{15}$$

where $\mathbf{I}_d$ is the $d \times d$ identity matrix, and $\mathbf{1}_m \in \mathbb{R}^m$ is a vector of ones.

The transition matrix $Q_{t|s} \in \mathbb{R}^{(d+m)\times(d+m)}$ is defined as:

$$Q_{t|s} = \alpha_{t|s}^{\leftarrow}\mathbf{I}_{(d,m)} + (1 - \alpha_{t|s}^{\leftarrow})\mathbf{M}_{(d,m)} + \boldsymbol{\pi}_{(d,m)} \tag{16}$$

which can be demonstrated intuitively:

$$Q_{t|s} = \begin{bmatrix} \alpha_{t|s}^{\leftarrow} & 0 & \cdots & 0 & \frac{1-\alpha_{t|s}^{\leftarrow}}{m} & \frac{1-\alpha_{t|s}^{\leftarrow}}{m} & \cdots & \frac{1-\alpha_{t|s}^{\leftarrow}}{m} \\ 0 & \alpha_{t|s}^{\leftarrow} & \cdots & 0 & \frac{1-\alpha_{t|s}^{\leftarrow}}{m} & \frac{1-\alpha_{t|s}^{\leftarrow}}{m} & \cdots & \frac{1-\alpha_{t|s}^{\leftarrow}}{m} \\ \vdots & \vdots & \ddots & \vdots & \vdots & \vdots & \ddots & \vdots \\ 0 & 0 & \cdots & \alpha_{t|s}^{\leftarrow} & \frac{1-\alpha_{t|s}^{\leftarrow}}{m} & \frac{1-\alpha_{t|s}^{\leftarrow}}{m} & \cdots & \frac{1-\alpha_{t|s}^{\leftarrow}}{m} \\ 0 & 0 & \cdots & 0 & \frac{1}{m} & \frac{1}{m} & \cdots & \frac{1}{m} \\ 0 & 0 & \cdots & 0 & \frac{1}{m} & \frac{1}{m} & \cdots & \frac{1}{m} \\ \vdots & \vdots & \ddots & \vdots & \vdots & \vdots & \ddots & \vdots \\ 0 & 0 & \cdots & 0 & \frac{1}{m} & \frac{1}{m} & \cdots & \frac{1}{m} \end{bmatrix} \begin{matrix} \\ \\ \\ \\ \\ \\ \\ \\ \end{matrix}$$

$$\underbrace{\phantom{xxxxxxxxxxxxxx}}_{\times d} \quad \underbrace{\phantom{xxxxxxxxxxxxxx}}_{\times m}$$

The corrupted data distribution is a direct derivative of Eq. 16 by setting $s = 0$:

$$\begin{aligned} x_t &= x_0 Q_{t|0} \\ &= \alpha_t x_0 \mathbf{I}_{(d,m)} + (1 - \alpha_t)x_0\mathbf{M}_{(d,m)} + x_0\boldsymbol{\pi}_{(d,m)} \\ &= \alpha_t x_0 + (1 - \alpha_t)x_0\mathbf{M}_{(d,m)} \\ &\sim \alpha_t x_0 + (1 - \alpha_t)\,\mathrm{U}(\mathcal{M}_m^L) \end{aligned} \tag{17}$$

where $\mathrm{U}(\mathcal{M}_m^L)$ is the uniform distribution on $\mathcal{M}_m^L$.

---

[4]To maintain simplicity, we use $\alpha_{t|s}^{\leftarrow} = \frac{\alpha_t}{\alpha_s}$ and $\alpha_{t|s}^{\rightarrow} = \frac{1-\alpha_s}{1-\alpha_t}$ to denote transition rate for the corruption and reverse process, respectively.

**Proof of the Reverse Process.** To generate a sequence of length $L$, the reverse process starts with $x_1 \sim U(\mathcal{M}_m^L)$. Let $\mathbf{a} \odot \mathbf{b}$ denote the Hadamard product between two vectors $\mathbf{a}$ and $\mathbf{b}$, the reverse process is inferred as:

$$
\begin{aligned}
q(x_s \mid x_t) &= \frac{Q_{t|s} x_t \odot Q_{s|0}^\top x_0}{x_t^\top Q_{t|0}^\top x_0} \quad \text{(D3PM deduction)} \\
&= \frac{[\alpha_{t|s}^\leftarrow \mathbf{I}_{(d,m)} x_t + (1 - \alpha_{t|s}^\leftarrow)\mathbf{M}_{(d,m)} x_t + \boldsymbol{\pi}_{(d,m)} x_t] \odot [\alpha_s x_0 + (1 - \alpha_s)\mathbf{M}_{(d,m)}^\top x_0]}{x_t^\top [\alpha_t x_0 + (1 - \alpha_t)\mathbf{M}_{(d,m)}^\top x_0 + \pi_{(d,m)}^\top x_0]} \\
&= \frac{[\alpha_{t|s}^\leftarrow \mathbf{I}_{(d,m)} x_t + (1 - \alpha_{t|s}^\leftarrow)\mathbf{M}_{(d,m)} x_t + \boldsymbol{\pi}_{(d,m)} x_t] \odot [\alpha_s x_0 + (1 - \alpha_s)\mathbf{M}_{(d,m)}^\top x_0]}{\alpha_t x_t^\top x_0 + (1 - \alpha_t) x_t^\top \mathbf{M}_{(d,m)}^\top x_0}
\end{aligned}
\tag{18}
$$

We consider the separate cases: $x_t^i = x_0^i$ and $x_t^i \in \mathcal{M}_m$.

**Case 1.** For $x_t^i = x_0^i$, Eq. 18 is simplified as

$$
\begin{aligned}
q(x_s^i \mid x_t^i = x_0^i) &= \frac{\alpha_{t|s}^\leftarrow x_0^i \odot \alpha_s x_0^i}{\alpha_t x_0^{i\,\top} x_0^i} \\
&= 1
\end{aligned}
\tag{19}
$$

**Case 2.** For $x_t^i \in \mathcal{M}_m$, we have

$$
\begin{aligned}
q(x_s^i \mid x_t^i \in \mathcal{M}_m) &= \frac{[(1 - \alpha_{t|s}^\leftarrow)\mathbf{M}_{(d,m)} x_t^i + \boldsymbol{\pi}_{(d,m)} x_t^i] \odot [\alpha_s x_0 + (1 - \alpha_s)\mathbf{M}_{(d,m)}^\top x_0]}{(1 - \alpha_t) x_t^{i\,\top} \mathbf{M}_{(d,m)}^\top x_0} \\
&= \frac{[(1 - \alpha_{t|s}^\leftarrow)\alpha_s \mathbf{M}_{(d,m)} x_t^i \odot x_0 + \boldsymbol{\pi}_{(d,m)}(1 - \alpha_s) x_t^i \odot \mathbf{M}_{(d,m)}^\top x_0]}{(1 - \alpha_t) x_t^{i\,\top} \mathbf{M}_{(d,m)}^\top x_0} \\
&= \frac{(\alpha_s - \alpha_t)\mathbf{M}_{(d,m)} x_t^i \odot x_0 + (1 - \alpha_s)\boldsymbol{\pi}_{(d,m)} x_t^i \odot \mathbf{M}_{(d,m)}^\top x_0}{(1 - \alpha_t) x_t^{i\,\top} \mathbf{M}_{(d,m)}^\top x_0}
\end{aligned}
\tag{20}
$$

Notice that $\alpha_{t|s}^\rightarrow = \frac{1 - \alpha_s}{1 - \alpha_t}$, and we have

$$
q(x_s^i \in \mathcal{M}_m \mid x_t^i \in \mathcal{M}_m) = \frac{1 - \alpha_s}{m(1 - \alpha_t)} = \frac{\alpha_{t|s}^\rightarrow}{m}
\tag{21}
$$

$$
q(x_s^i \notin \mathcal{M}_m \mid x_t^i \in \mathcal{M}_m) = \frac{\alpha_s - \alpha_t}{1 - \alpha_t} = 1 - \alpha_{t|s}^\rightarrow
\tag{22}
$$

Combining case 1 with case 2, we have

$$
q(x_s^i \mid x_t^i) = \begin{cases}
1, & \text{if } x_s^i = x_t^i, \ x_t^i \notin \mathcal{M}_m, \\
\alpha_{t|s}^\rightarrow / m, & \text{if } x_s^i \in \mathcal{M}_m, \ x_t^i \in \mathcal{M}_m, \\
1 - \alpha_{t|s}^\rightarrow, & \text{if } x_s^i \notin \mathcal{M}_m, \ x_t^i \in \mathcal{M}_m, \\
0, & \text{otherwise.}
\end{cases}
\tag{23}
$$

Following MDLM's deduction, assume that the denoising network can reconstruct $x_0$ perfectly, we use $p_\theta(x_t)$ to approximate this reverse process for complex sequences, and get

$$
q(x_s^i \mid x_t) = \begin{cases}
1, & \text{if } x_s^i = x_t^i, \ x_t^i \notin \mathcal{M}_m, \\
\alpha_{t|s}^\rightarrow / m, & \text{if } x_s^i \in \mathcal{M}_m, \ x_t^i \in \mathcal{M}_m, \\
(1 - \alpha_{t|s}^\rightarrow) p_\theta^i(x_t), & \text{if } x_s^i \notin \mathcal{M}_m, \ x_t^i \in \mathcal{M}_m, \\
0, & \text{otherwise.}
\end{cases}
\tag{24}
$$

## C    COMPLEXITY AND LIMITATIONS

Our rehash sampler follows DDM-theoretic principles and is implemented using torch.multinomial, which internally relies on Gumbel-max (Zheng et al., 2024). As a result, its asymptotic computational complexity is comparable to that of the original MaskGIT sampler. Practical efficiency gains, however, stem from two key factors. First, discrete visual tokens exhibit high correlation and redundancy, allowing multiple tokens to be predicted simultaneously and reducing the number of required operations compared with fully autoregressive or diffusion-based approaches. Second, DDM decouples the scheduler during training and sampling (Sahoo et al., 2024), enabling high-quality reconstruction from noise using arbitrarily defined timesteps. Using a cosine scheduler, ReDDiT achieves improved sample quality within fewer steps, as validated empirically.

Despite its effectiveness, the rehash sampler has limitations that motivate future work. The current rehashing strategy is applied at every step, but the impact on diversity and convergence is not fully characterized, suggesting that optimized rehash frequency or intensity could improve performance. Additionally, integrating ReDDiT with dynamic token-update mechanisms in DFM frameworks may further enhance sample quality and diversity, particularly for complex multimodal generation tasks. These considerations highlight potential directions for extending and refining discrete diffusion sampling methods.

## D    SAMPLING FROM LEARNED NETWORKS

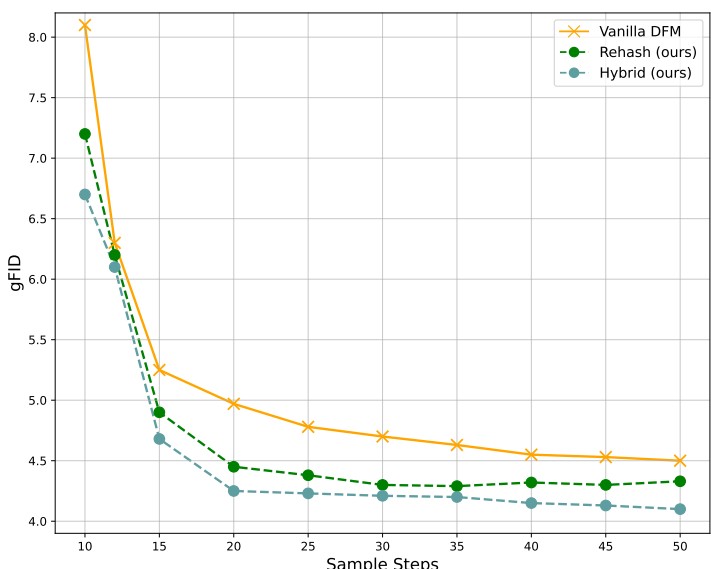

Figure 7: Generation quality comparison with DFM methods. The experiments are conducted on ReDDiT-L with a constant classifier-free guidance (cfg = 2.0).

We present a detailed version of discrete flow matching (DFM) sampler 3, and discuss the integration of it with ours. Fig. 7 presents a quantitative comparison of the vanilla DFM sampler, our proposed rehash sampler, and a hybrid strategy that combines both approaches by incorporating selected DFM steps into the rehash trajectory. All methods are evaluated using identical model weights, as the training objectives are compatible due to their shared time-weighted loss formulation.

The rehash sampler exhibits stronger overall performance than DFM, especially in the 15–32 step range, where it achieves low and stable gFID scores. This suggests that our modification enables more efficient decoding trajectories without sacrificing sample quality. The hybrid variant, which

---

**Algorithm 3** DFM Sampling Stepwise Pseudo Code

---

**Require:** $x_t$, labels, timestep $t$, step size $\Delta t$
  1: Compute jump probabilities: $j_t \leftarrow 1 - \alpha_t$, $j_s \leftarrow 1 - \alpha_{t-\Delta t}$
  2: Determine guidance scale $\omega$ from schedule
  3: Obtain logits $\text{logits}_{\text{cond}}$, $\text{logits}_{\text{uncond}}$ via forward pass
  4: $\text{logits}_{x_0} \leftarrow \text{logits}_{\text{uncond}} + \omega \cdot (\text{logits}_{\text{cond}} - \text{logits}_{\text{uncond}})$
  5: $p_{x_0} \leftarrow \texttt{softmax}(\text{logits}_{x_0})$
  6: Sample $\hat{x}_0 \sim p_{x_0}$ using categorical sampling
  7: Construct one-hot encodings: $\delta_{x_0}, \delta_{x_t}$
  8: corrective $\leftarrow \frac{j_s}{j_t} \cdot \delta_{x_t}$
  9: $u \leftarrow \frac{j_t - j_s}{j_t} \cdot \delta_{x_0}$
 10: Overwrite $u$ in masked range with corrective terms
 11: Mask entries already present in $x_t$ from $u$
 12: Compute total transition intensity: $\lambda \leftarrow \sum u$, elementwise
 13: Draw Bernoulli mask: $M \sim \text{Bernoulli}(1 - \exp(-\lambda))$
 14: For each masked position in $M$, sample from categorical $u$ to obtain updated $x_s$
 15: **return** $x_s$

---

integrates only the middle and final steps of the DFM update into the rehash schedule, also delivers consistent gains over the vanilla DFM, suggesting that partial refinement from DFM is beneficial even when the majority of the trajectory is governed by our rehash dynamics.

By leveraging shared gradual decoding infrastructure, the hybrid approach enables practical integration of DFM refinement into the ReDDiT framework with minimal overhead. As noted in the main paper, this leads to a $\sim$0.1 improvement in gFID on ImageNet-1K, reinforcing the complementary strengths of the two samplers. We leave the comprehensive study on the optimal integration of different samplers for future exploration.

## E  EXPERIMENT DETAILS

We provide detailed training and generation configurations for ReDDiT in Table 5. Our method incorporates DINOv2-B for representation alignment, which requires computing image features during the forward pass (only activated during training). This introduces an overhead, making training roughly 1.2× slower than solely on discrete tokens. However, this additional cost is offset by faster convergence and improved stability, particularly in early training stages.

The use of quantized latents allows for larger batch sizes under limited GPU memory, making our approach more accessible for low-resource settings. Additionally, aligning discrete codes with semantic features improves the quality and diversity of learned representations. Overall, our design balances computational efficiency with model performance, making it a practical choice for both research and deployment.

## F  DISCUSSION ON REPRESENTATION ALIGNMENT

Representation Alignment (RepA) introduces a similarity-based auxiliary loss that aligns intermediate features of the diffusion model with pretrained DINOv2 embeddings. Although originally proposed for continuous diffusion, its effectiveness naturally extends to the discrete setting. In ReDDiT, discrete tokens are first mapped into a continuous embedding space, after which the architecture is identical to transformer-based continuous diffusion models (e.g., DiT). This means that the absence of continuous input does not fundamentally alter the structure of the model's internal representations. However, discrete tokenizers restrict direct gradient flow from pixels to the codebook, making it more difficult for the model to organize high-level semantics during early training.

RepA provides an external semantic scaffold that compensates for this difficulty. By encouraging the network to match DINOv2's robust visual features, RepA helps establish meaningful structure in the latent representations before the denoising objective becomes sufficiently informative. Empirically,

Table 5: Experiment details for ReDDiT on ImageNet-1K. *Vari.* refers to a time-variant growing guidance scale following MDTv2, which is a common practice for diffusion models.

| Setting | ReDDiT-L (Ablation) | ReDDiT-L | ReDDiT-XL | ReDDiT-XL$_{f8}$ |
|---|---|---|---|---|
| Hidden Size | 1024 | 1024 | 1280 | 1280 |
| Transformer Block | 24 | 24 | 28 | 28 |
| Attention Head | 16 | 16 | 20 | 20 |
| Image Tokenizer | LlamaGen-f16 | IBQ-f16 | IBQ-f16 | LlamaGen-f8 |
| Codebook Size | 16384 | 16384 | 16384 | 16384 |
| Noise Capacity | 128 | 1024 | 1024 | 128 |
| Sequence Length | 256 | 256 | 256 | 1024 |
| RepA Latent Size | 16×16 | 16×16 | 16×16 | 32×32 |
| Batch Size | 64 | 64 | 32 | 16 |
| Global Batch Size | 1024 | 1024 | 1024 | 1024 |
| LR scheduler | Cosine Decay | Cosine Decay | Cosine Decay | Cosine Decay |
| Learning Rate | 3e-4 | 3e-4 | 3e-4 | 4e-4 |
| Minimal LR | 1e-5 | 1e-5 | 1e-5 | 1e-5 |
| Warmup Steps | 2k | 2k | 2k | 2k |
| Training Steps | 500k | 500k | 500k | 500k |
| Training Time | ∼1 day | ∼1 day | ∼2 days | ∼3 days |
| Generation CFG (*Vari.*) | 1.0-5.0 | 1.0-6.5 | 1.0-6.5 | 1.0-5.5 |

Table 6: **Acceleration of ReDDiT using response cache $K_r$.**

| Model | Config | | Performance | |
|---|---|---|---|---|
| | Steps | $K_r$ | Relative Speed | gFID↓ |
| ReDDiT-L | 32 | 2 | ×1.33 | 2.28 ($\Delta = 0.15$) |
| ReDDiT-XL | | | ×1.52 | 1.88 ($\Delta = 0.14$) |
| ReDDiT-XL | 64 | 4 | ×2.17 | 1.83 ($\Delta = 0.09$) |
| ReDDiT-XL$_{f8}$ | | | ×2.56 | 1.71 ($\Delta = 0.10$) |

we observe that removing RepA leads to an early training plateau, whereas with RepA the alignment loss rapidly decreases and stabilizes, indicating improved organization of semantic information. This behavior mirrors observations in continuous diffusion models and supports the view that RepA offers a general mechanism for accelerating convergence, independent of whether the base diffusion process is discrete or continuous.

## G  ACCELERATING REDDiT

Recent efforts on scaling and accelerating discrete diffusion models are making this generative paradigm more practical than theoretical attempts. We adapt the dLLM-Cache Liu et al. (2025) design into our framework, which efficiently reuses intermediate computations without compromising model performance. Since the condition is modulated using AdaLN and introduces minimal calculation, we do not activate $K_p$ (cache for prompt). As the decoding of visual sequence varies with time more quickly than language decoding, we implement the cache for response with small values like $K_r = 2$ or $4$, which means the $K$ and $V$ of transformer layer is updated every 2 or 4 decoding steps instead of per step. As shown in Tab. 6, the inference speed is boosted up to 2 times with minimal performance drop, which makes our largest model ReDDiT-XL$_{f8}$ comparable to diffusion models with accelerated solvers.

## H  QUALITATIVE RESULTS

We provide more samples of ReDDiT's generation in Fig. 8.

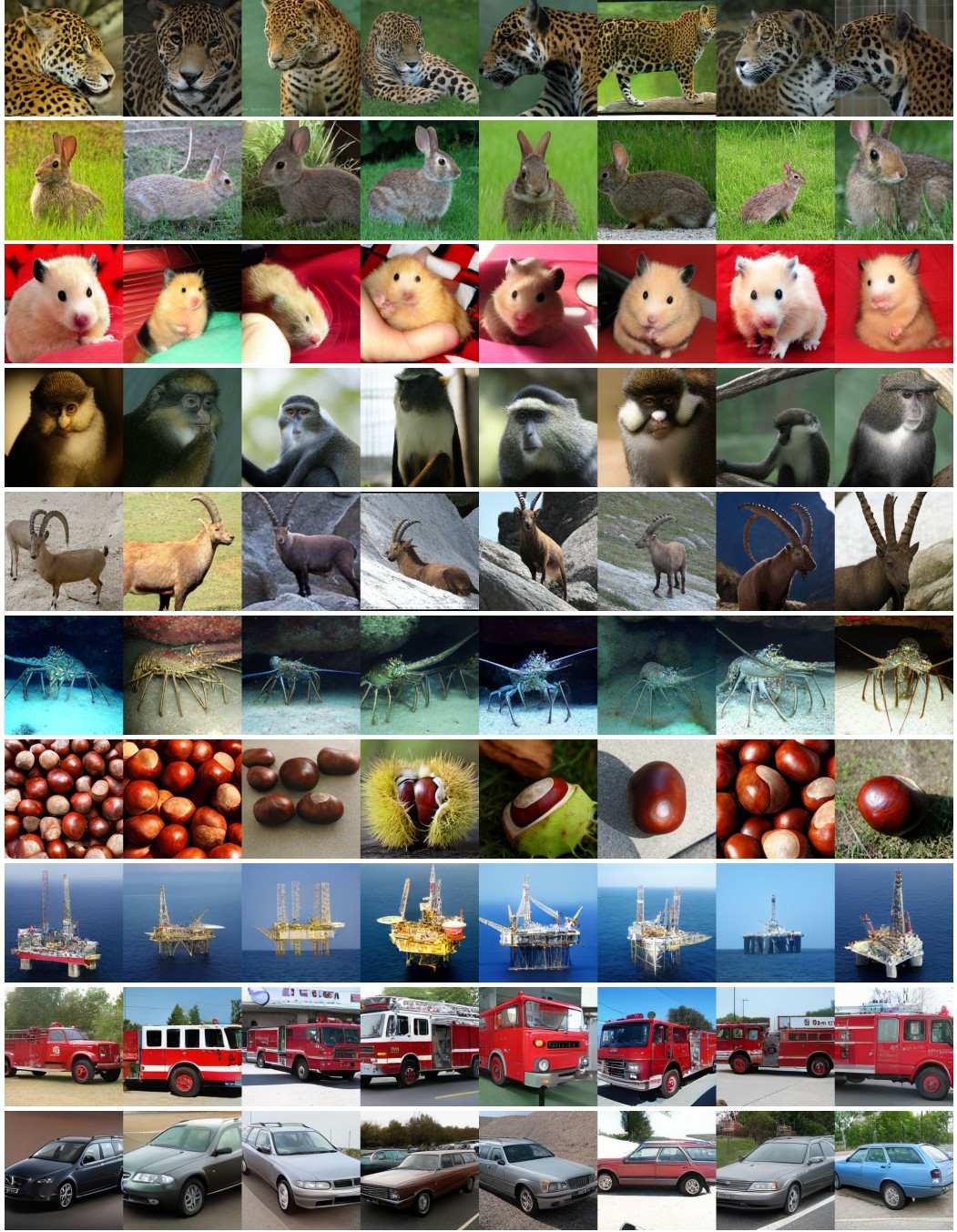

Figure 8: Class-conditional generation samples of ReDDiT on ImageNet $256 \times 256$.

