# OpenReview forum: "ReDDiT: Rehashing Noise for Discrete Visual Generation"
_ICLR.cc/2026/Conference — ICLR 2026 Poster_

### Official Review · Reviewer_MWhn · 2025-10-29

**Soundness:** 4
**Presentation:** 4
**Contribution:** 3
**Rating:** 8
**Confidence:** 3

**Summary:**

This paper tackles a central problem of discrete diffusion models, especially for the masked visual token models (MVTMs), which have relied on a single absorbing state to perturb the input discrete visual token and Gumbel softmax with relatively high sampling temperature. Existing sampling strategies introduce instability and degradation as vocabulary size grows. By introducing masking capacity, ReDDiT effectively improves the capabilities of MVTMs. The experimental results are promising.

**Strengths:**

- **Well-Motivated and Simple-yet-General Solution**: While discrete diffusion models (DDMs) mimic the continuous one, a single absorbing state is a fundamental restriction of the MVTMs, which limits the diversity of samples and enforces MVTMs to utilize high sampling temperature, which compromises sample quality.  By simply extending the absorbing state design, ReDDiT achieves promising results, paving the way for improving DDMs' generative capabilities.

- **Clear Presentation**: The visualization in Figs 1 and 3 is good. The paper is well-structured, facilitating a clear understanding of the background and contributions of the paper. Simple notation with a theoretically grounded explanation ensures the readability and accessibility of the paper.

- **Generalizability and Significance**: The proposed rehashing masking can be generally adopted by the masked generative modeling paradigm, bringing a significant advancement to the field.

**Weaknesses:**

- **Controlling the Stochasticity**: Controlling the stochasticity of the sampling process is critical, as some tasks, such as drug design, require exact outputs at the expense of diversity, while in some cases, like artistic painting, require high variance. However, the proposed method does not utilize sampling temperature, lacks such control. This may significantly limit the applicability of the method in various fields. I think the proposed method could also adopt a sampling temperature. In some research, sampling with high temperature equipped with an appropriate guidance scale could show superior sample quality and diversity simultaneously [1]. I think it is more beneficial if the stochastic sampling can also be adopted for the ReDDiT sampling pipeline.

- **Justification for Exact Sampling in Large Vocabulary**: The authors argue that ReDDiT captures the distribution better with a large vocabulary size. However, I can't find any theoretical justification for this statement. I agree that the noise rehashing can express better diversity without heuristic temperature scaling. However, I wonder why it can capture the exact distribution under large vocabulary settings.

- **Scalability to Sampling Steps**: In Table 1, the optimal sampling steps increase as the models' capacity grows. MaskGIT has shown the "sweet spot" of sampling steps, where large sampling can degrade the sample quality, conversely. The scalability analysis of the ReDDiT sampler with respect to the number of sampling steps is insufficient. The plot in Figure 3, showing gFID versus sampling steps, is hard to interpret, does not indicate the sampling steps, and fails to demonstrate a clear trend.

[1]  Unlocking the Capabilities of Masked Generative Models for Image Synthesis via Self-Guidance, NeurIPS 2024

**Questions:**

- Low sampling temperature is known to incur a multi-modality problem in non-autoregressive (NAR) sampling since NAR samples multiple tokens at a time, and their relationship is ignored. Since ReDDiT does not utilize sampling temperature and uses multinomial sampling, I wonder if ReDDiT also suffers from such a limitation, especially for the small or large sampling steps.

- The rehash operation (Algorithm 2, line 6) is mentioned briefly. This step randomly shuffles tokens that are already noise tokens at the start of each sampling step. What is the theoretical and empirical justification for this? What happens to performance (gFID, diversity) if this line is removed? (i.e., what happens if the sampling becomes deterministic to the initial noise?)

**Details Of Ethics Concerns:**

No ethical concerns raised.

---

> ### Author Response · Authors · 2025-11-24
>
> Thank you for the positive feedback. We appreciate your recognition of our well-motivated and general solution, the clarity of presentation, and the potential generalizability and significance of the proposed rehashing masking approach. We respond to the questions point-to-point below.
> ___
>
> **Q1 (w1, combined with question-1): On stochastic sampling and temperature scheduling**
>
> **A1:** Thank you for highlighting this line of work. We are aware that several discrete-generation systems combine temperature scaling with guidance to jointly improve quality and diversity. For example, the paper you mentioned, as well as practical implementations such as TiTok[1]  (which mixes MaskGIT-style sampling with temperature and noise-strength control). Motivated by these observations, we experimented with incorporating temperature into the ReDDiT sampling pipeline and summarize our findings here.
>
> **Using MaskGIT-style sampling.**
>
> When paired with large-vocabulary visual tokenizers like llamagen-f16, MaskGIT sampling is extremely sensitive to temperature. Although temperature tuning can improve visual diversity, the final metrics remained consistently worse than our multinomial-based sampler. We attribute this to its weaker compatibility with large-token vocabulary models: as shown in Fig. 3, even small changes in the temperature schedule create large accuracy–diversity tradeoffs, and no schedule generalizes across LlamaGen, IBQ, and MaskGIT tokenizers, as we experimented. Due to the instability and tokenizer-specific tuning burden, we did not pursue this direction further.
>
> **Using multinomial sampling with temperature.**
>
> Temperature scaling can also be applied to logits before softmax and multinomial sampling. This is particularly helpful at very small sampling steps. For example, with ReDDiT-XL at 16 sampling steps, a decreasing temperature schedule (initially promoting diversity, later suppressing it) improves FID by ~0.5 (2.53→2.08) and IS by ~20 (273→291). However, because we already employ a dynamic CFG schedule, its interaction with temperature becomes highly coupled, requiring substantial grid search to find optimal settings. For clarity and reproducibility, we therefore use a constant temperature in the main paper and rely on CFG scaling to navigate the quality–diversity trade-off, which is the most standard practice.
>
> We agree that stochasticity control remains an important and under-explored dimension for discrete diffusion, and have included the work you mentioned into the main paper. We hope that future work could conduct more systematic studies on the sampling issues of DDMs.
>
> [1] Yu et al. An Image is Worth 32 Tokens for Reconstruction and Generation. NeurIPS 2024.
>
> ___
>
> **Q2 (w2). Under large vocabulary, why ReDDiT captures the distribution better than previous methods?**
>
> **A2:** Good question! Prior work such as [2] (from the language-modeling perspective) provides valuable intuition: the precision of Gumbel noise, the overly restrictive top-k/top-p–like decoding behaviors, and the high-entropy nature of modern learned visual tokens all affect the quality of token-level sampling.
>
> Empirically, we observe a clear distinction:
>
> * With a small vocabulary (e.g., MaskGIT’s native 1,024-token tokenizer), sampling with Gumbel-max and multinomial yields similar performance (≤0.4 FID difference).
> * With large vocabularies (e.g., LlamaGen-f8), the gap becomes substantial (>1 FID difference; see Fig. 3).
>
> This suggests that large vocabularies amplify the limitations of deterministic or low-precision sampling mechanisms. Our pipeline mitigates this by:
>
> * using multinomial sampling to avoid Gumbel precision issues, and
> * using rehash noise to maintain sampling diversity across steps.
>
> [2] Zheng et al. Masked Diffusion Models are Secretly Time-Agnostic Masked Models and Exploit Inaccurate Categorical Sampling. ICLR 2025
> ___
>
> **Q3 (w3). Scalability with sampling steps**
>
> **A3:** Thank you for pointing this out. Classical MaskGIT indeed shows a clear “sweet spot” , partly because its sampling is tightly coupled to its training objective through cosine masking schedules. However, modern DDM formulations largely decouple training and sampling, and makes the schedule less sensitive.
>
> In our experiments, once the number of sampling steps reaches a moderate range, the performance plateaus: quality does not degrade significantly even when the number of steps increases further. This is consistent with what has been observed in DFM[3]. Appendix Fig. 7 visualizes this behavior.
>
> [3]Gat et al. Discrete Flow Matching. NeurIPS 2024

---

> > ### Author Response · Authors · 2025-11-24
> >
> > **Q4 (question-2): Justification for the rehash operation**
> >
> > **A4:** Thank you for raising this question. The rehash step is designed to encourage path diversity during generation by resampling positions that are already noise tokens. We ran controlled experiments using models trained with ( m=1 ) and ( m=128 ), then modifying the sampler in two ways:
> >
> > (1) fixing the absorbing state to the first token, and
> >
> > (2) disabling rehash after random initialization.
> >
> > A summary of the results is shown below:
> >
> > | Setting | gFID | Notes |
> > |---|---|---|
> > | m=1 (baseline) | 4.13 | – |
> > | m=128,fixed absorbing state=1 | 4.25 | Worse than m=1 despite larger noise capacity |
> > | m=128, no rehash after init | 4.07 | Better but not full performance |
> > | m=128, rehash enabled | 3.92 | Fully unlocks model capacity |
> >
> > The results indicate that without rehash, the model cannot fully exploit the benefits of large noise capacity, and sampling becomes overly deterministic. **We have included these results and a short discussion in the main paper Section 3.4**.
> > ___
> >
> > We hope these clarifications address your concerns and are helpful for your final assessment. Thank you again for the thoughtful and detailed feedback.

---

> > ### Comment · Reviewer_MWhn · 2025-11-27
> >
> > Thank you for the detailed response. I would like the authors to complete Figure 7 with sampling steps of 45 and 50 for the comprehensive experiment. Given that the weaknesses are minor issues, I keep my positive ratings.

---

> > > ### Author Response · Authors · 2025-11-27
> > >
> > > Thanks for your positive feedback! We’re glad to hear that the our response helped address your concerns.
> > >
> > > We have updated Figure 7 with more data points in step range [10,12,15,20,25,30,35,40,45,50].

---

### Official Review · Reviewer_DAHj · 2025-10-31

**Soundness:** 2
**Presentation:** 3
**Contribution:** 3
**Rating:** 6
**Confidence:** 3

**Summary:**

The paper proposed a new rehashing noise approach for discrete diffusion transformer, which guarantees high diversity  and low discrepancy of the generation, outperforming baseline model by a large margin. Supported by solid theory and extensive experiments, the effectiveness and advantages of the method have been validated. The overall quality is satisfactory, while more explanation and interpretation in methodology are needed for clear presentation.

**Strengths:**

1. The paper proposed a new sampling methods that facilitate efficient and diverse generation for discrete diffusion, and it has a good structure with clear motivation, distinctive contributions, and solid theory.
2. The proposed method is well-supported by authors’ experiment results, either in tables other than figures.

**Weaknesses:**

1. The high diversity of the generation can be seen from qualitative examples, while what does low discrepancy mean in the proposed sampler?
2. The methodology part is somewhat confusing. For equation 6, the definition of \mathbf{m}_j is not clear. Does \mathbf{m}_j indicate a absorbing token at the j-th position of the vector? Plus, I=0 and j=0 does mot make sense for index starting from 1. For equation 8, after rewriting equation 1, why does the transition kernel become \frac{1}{m}? If it is due to m states of  \mathcal{M}, then this problem comes back again to the definition of \mathbf{m}\in\\mathbb{R}^{m}: does \mathbf{m}_{j} indicate m token at position j and elsewhere all 0? Is there only one absorbing state, which is m? Moreover, for equation 9, the symbol usage is not consistent, sometime it is $x_{t}^{I}$,  while sometimes it is $x_{t}$.
3. More interpretation needed for figures. For fig 2, it is not clear how the learned distributions is better than ordinal ones. Is it because of a hyperplane between visual vocabulary and the rehashing noise?
4. The author tested on well-known ImageNet dataset, while the performance generalization to other (more challenging) datasets is not clear.

**Questions:**

1. Can authors analyze the complexity of ReDDiT sampler? Does the efficiency come from less operation needed?
2. The discussion of the limitations of the proposed sampler  and how to improve it would be beneficial for future work of researchers.

---

> ### Author Response · Authors · 2025-11-24
>
> Thank you for the encouraging feedback. We appreciate your recognition of the clear motivation, solid theoretical foundation, and the strong experimental support for our proposed sampling method. We carefully respond to your questions point-to-point below.
>
> ___
>
> **Q1 (w1). Meaning of “low discrepancy” in the proposed sampler**
>
> **A1:** Thank you for this excellent question. By low discrepancy, we refer to the improved sampling accuracy of our sampler—specifically, its ability to better match the model’s predicted token distribution.
>
> Since discrete diffusion must sample a token at every denoising step, accurate categorical sampling is crucial. Prior samplers rely on explicit score-based Gumbel-max tricks, whose numerical precision is limited due to floating-point instability in max-operations over large vocabularies. Please refer to figure 3 for the demonstration of vanilla Gumbel-max's effect on sample distributions, and [1] for a detailed technical discussion.
>
> In the rehash sampler, we replace Gumbel-max with torch.multinomial, which performs direct categorical sampling using the full probability vector. Meanwhile, the extended noise vocabulary is also carefully handled to avoid  truncation on small values. This reduces the discrepancy between the true model distribution and the sampled trajectories, producing more faithful generations without altering diversity. We have updated Section 2.2 in the main paper for clarity.
>
> [1] Zheng et al. Masked Diffusion Models are Secretly Time-Agnostic Masked Models and Exploit Inaccurate Categorical Sampling. ICLR 2025
>
> ____
>
> **Q2 (w2-1). Clarification of notation in Equation (6)**
>
> **A2:** Thank you for carefully checking the notation.
> The full vocabulary in ReDDiT is
> {$e_0,e_1,…,e_d,m_1,m_2,…,m_m$}
> where $e_i$ are visual tokens and $m_j$ are mask/noise tokens. Because E and M are two disjoint basis spaces, we use the tuple notation $v_{(i,j)}$ to indicate from which space a token originates.
>
> * i=0 indicates that the token comes purely from the mask space.
> * j=0 indicates that the token comes purely from the visual space.
> * i and j are never simultaneously zero.
>
> We have revised the main text to make this clearer and avoid any confusion.
>
> ___
>
> **Q3 (w2-2). Clarification of Equation (8): why the transition kernel becomes 1/m**
>
> **A3:** We indeed introduce m absorbing states, corresponding to m distinct mask tokens. If a token is already masked, it remains in the mask space during subsequent transitions. Because all mask absorbing states are equally likely under the base kernel, the transition probability to any specific absorbing state becomes 1/m. We have clarified this point and improved the accompanying explanation.
> ___
>
> **Q4 (w2-3). Notation inconsistency in Equation (9)**
>
> **A4:** Thank you for your meticulous review. Following Section 2.1, superscript i refers to transitions for the i-th token. The DDM theory derives a per-token ELBO approximation that lifts to the full sequence. Due to space constraints, we skipped some intermediate steps, which caused inconsistent notation. We have modified the symbols for clarity.
> ___
>
> **Q5 (w3). More interpretation for Figure 2**
>
> **A5:** Thank you for raising this point. The key intuition is as follows:
>
> * In the ideal high-dimensional embedding space, the learned noise distribution shall be close to isotropic, forming a smooth distribution around or inside the visual vocabulary manifold.
> * However, t-SNE (used for 3D visualization) emphasizes local structure and often exaggerates cluster separation. Thus, an isotropic distribution in 1024D can appear directional or distorted after projection.
> * Additionally, the visual vocabulary itself is not isotropic; embeddings from VQ tokenizers often form irregular clusters rather than spherical ones.
>
> Therefore, while Figure 2 may not visually show isotropy, it still reveals the expected separation between noise tokens and visual tokens in the model’s learned embedding. We have modified the caption accordingly.

---

> ### Author Response · Authors · 2025-11-24
>
> **Q6 (w4). Generalization to more challenging datasets**
>
> **A6:**  Thank you for this insightful question.
> ImageNet-1K remains the standard benchmark for evaluating generative diffusion models (e.g., DiT, U-ViT). Scaling ReDDiT to substantially larger datasets is computationally expensive within the rebuttal period. However, we conducted stress tests under harder semantic conditions (reduced AdaLN capacity and in-context conditioning), and ReDDiT continues to show consistent improvements. These results suggest that the method remains robust under more complex semantic distributions. (As this question mirrors Reviewer daQc’s concern, please refer to the response for further details. )
>
> | Method| Capacity | #Parameters | gFID $\downarrow$ | IS $\uparrow$ |
> |---|---|---|---|---|
> | ReDDiT-shrink | m=1 |  270M | 3.86 | 260.1 |
> | ReDDiT-shrink | m=128 | 270M | 3.46 | 272.1 |
> | ReDDiT-context | m=1 | 229M | 4.01 | 241.4 |
> | ReDDiT-context | m=128 | 229M | 3.71 | 254.9 |
>
> Across both settings, ReDDiT consistently improves generative quality, demonstrating that rehashed noise remains beneficial even when semantic information is highly compressed or weakly provided. These experiments strongly suggest that the mechanism is compatible with harder distributions regimes.
>
> ___
>
> **Q7 (question-1). Can authors analyze the complexity of ReDDiT sampler? Does the efficiency come from less operation needed?**
>
> **A7:** Thank you for the question. The ReDDiT sampler (i.e., the rehash sampler) is strictly designed following DDM theory and implemented using torch.multinomial. Since its underlying implementation still relies on Gumbel-max, its computational complexity is not fundamentally different from the original MaskGIT sampler (see discussion in [1]). However, by avoiding the forced decoding of all tokens and incorporating the rehash operation, ReDDiT alleviates certain constraints during DDM sampling.
>
> We believe the efficiency primarily arises from two aspects:
>
> 1. Discrete visual tokens are highly correlated and redundant. Predicting multiple tokens simultaneously is inherently faster than diffusion or autoregressive generation. MaskGIT, as a precursor of DDM, already demonstrated faster sampling than contemporary continuous diffusion methods despite limited performance.
> 2. Compared with MaskGIT, DDM decouples the scheduler during training and sampling. This is analogous to the transition from DDPM to DDIM, allowing the model to reconstruct a complete image from noise in arbitrarily defined timesteps. Experiments (Table 3b) show that using a cosine scheduler achieves the best sampling quality within a given number of steps. Recent work [2] even shows that, in some settings, a cosine scheduler yields theoretically optimal results for DDM.
>
> [1] Zheng et al. Masked Diffusion Models are Secretly Time-Agnostic Masked Models and Exploit Inaccurate Categorical Sampling. ICLR 2025
>
> [2] Zhang et al. The Cosine Schedule is Fisher-Rao-Optimal for Masked Discrete Diffusion Models.
>
> ___
>
> **Q8 (question-2). The discussion of the limitations of the proposed sampler and how to improve it would be beneficial for future work of researchers.**
>
> **A8:** Thank you for the suggestion. The ReDDiT sampler has the following limitations:
> 1. Rehash at every step: Although rehashing at every step is theoretically valid, its precise impact on diversity is not fully understood. Investigating the effects of varying rehash intensity and sampling temperature is a promising direction for future research.
> 2. Combination with token-update DFM mechanisms: Our paper proposes integrating ReDDiT with such mechanisms, which also presents a potential avenue for further improvement.
>
> **We have incorporated the discussion from Q7 along with these limitations into the Appendix C** and will integrate them into the main text in the camera-ready version.
>
>
> ___
>
> We hope the above clarification addresses your concerns and is helpful for your final assessment. Thank you again for the thoughtful feedback.

---

### Official Review · Reviewer_gKak · 2025-11-01

**Soundness:** 4
**Presentation:** 3
**Contribution:** 3
**Rating:** 6
**Confidence:** 3

**Summary:**

This paper introduces ReDDiT (Rehashing Noise for Discrete Diffusion Transformer), a novel discrete diffusion framework for high-quality visual generation. The authors identify two fundamental limitations in existing
masked visual token models (MVTMs): (1) the use of a
single absorbing (mask) token, which restricts the expressivity of the noise process, and (2) reliance on
heuristic, Gumbel-max–based sampling, which is unstable and requires heavy tuning—especially with large codebooks. To address these, ReDDiT proposes
rehashing noise, which expands the absorbing state into
multiple randomized indices, thereby enriching the diffusion trajectory space. It further introduces a
principled rehash sampler grounded in discrete diffusion theory, which uses multinomial sampling over softmax probabilities to ensure high diversity and low discrepancy without ad-hoc randomness. On ImageNet 256×256, ReDDiT
achieves a gFID of 1.61, significantly outperforming prior discrete models (e.g., MaskGIT: 6.18) and approaching the performance of top continuous diffusion models, while maintaining superior inference efficiency.

**Strengths:**

1.Significant empirical improvement: ReDDiT delivers a substantial leap in generation quality for discrete models, closing much of the gap with continuous diffusion while preserving the efficiency advantages of discrete token-based generation.
2. Insightful problem diagnosis and elegant solution: The paper clearly articulates the shortcomings of single-mask noise and Gumbel-based sampling, and the proposed rehashing noise mechanism is both theoretically motivated and practically effective.
3. Clear presentation and thorough evaluation: The writing is accessible, and the paper includes comprehensive ablations, visualizations (e.g., Figure 1, 3), and comparisons that convincingly demonstrate the contribution of each component.

**Weaknesses:**

1.Minor performance gap with best continuous models: While ReDDiT achieves gFID = 1.61, the best continuous models (e.g., MDTv2) report gFID ≈ 1.58 under similar settings. Although the efficiency advantage is compelling, the paper could more explicitly acknowledge this small but notable gap as a limitation or future direction.
2. Tokenizer-dependent hyperparameter tuning: The optimal noise capacity m varies with the tokenizer (e.g., m=128 for LlamaGen-f8 vs. m=1024 for IBQ), requiring empirical tuning. This slightly undermines the method’s plug-and-play appeal and generalizability across tokenization schemes.

**Questions:**

1.The paper successfully adapts Representation Alignment (RepA) from continuous to discrete diffusion. It would strengthen the work to include a brief discussion (even in the appendix) on why aligning discrete token embeddings with continuous DINOv2 features is effective—e.g., whether it stabilizes training by providing semantic priors in the absence of gradient flow through the tokenizer.
2. In Figure 4(a), the x-axis should be labeled “Training Steps” (or similar) for clarity.

---

> ### Author Response · Authors · 2025-11-24
>
> Thank you for the positive evaluation. We appreciate your recognition of the strong empirical improvements, the clarity of our problem diagnosis and solution, and the thoroughness of our presentation and evaluations. We carefully respond to your questions point-to-point below.
> ___
>
> **1. Why RepA remains effective when adapted to discrete diffusion**
>
> We appreciate the reviewer highlighting this important aspect. Representation Alignment (RepA) plays a central role in accelerating convergence, and its extension from continuous to discrete diffusion is indeed non-trivial.
> Although ReDDiT operates on quantized latent codes, its architecture closely mirrors that of DiT: after token embedding, all subsequent processing occurs in a continuous latent space through transformer blocks. Thus, the quantization step affects only the input modality, not the internal representation dynamics.
>
> This structural similarity explains why aligning intermediate features with DINOv2 embeddings remains effective. In discrete diffusion, the absence of gradient flow through the tokenizer limits the model’s ability to learn meaningful semantic structure early in training. RepA compensates for this by providing a strong, pretrained semantic prior, guiding the network toward well-structured representations before the denoising objective alone becomes sufficiently informative.
>
> To further validate this intuition, we performed an ablation study where RepA is removed, and training exhibits an early-stage plateau, suggesting difficulty in organizing discrete token embeddings without semantic scaffolding. With RepA, the alignment loss decreases rapidly and stabilizes at a low value, consistent with observations in continuous diffusion models.
>
> **We have added a concise discussion of this rationale to Appendix F, following your suggestion.**
> ___
>
> **2. Tokenizer-dependent tuning of noise capacity**
>
> We agree with the reviewer that the optimal noise capacity (m) varies across tokenizers (e.g., (m=128) for LlamaGen-f8 vs. (m{=}1024) for IBQ). This reflects an inherent difference in how much “diversity regularization” each tokenizer requires: tokenizers with coarser quantization or more heterogeneous codebooks benefit from stronger negative-sample regularization.
>
> While this introduces a small amount of empirical tuning, we note that:
>
> * The search space is low-dimensional (typically $m \in \[16, 128\]$).
> * The optimal region is broad; performance is stable as long as (m) is neither too small nor excessively large.
> * Once chosen, (m) remains fixed for the entire training and sampling process.
>
> We consider exploring tokenizer-adaptive or learned noise capacity schedules an interesting direction for future work, and **have added a discussion in Appendix C.**
>
> ___
>
> **3. Minor: Small remaining gap with best continuous models**
>
> Thank you for pointing this out. We now explicitly acknowledge the top-tier continuous diffusion counterparts in the revised version.
> We view closing this narrow gap as an important future direction, particularly as discrete diffusion continues to improve in efficiency and scalability. We have also corrected the labels of figures for clarity.
> ___
>
> We hope the above clarification addresses your concerns and is helpful for your final assessment. Thank you again for the thoughtful feedback.

---

### Official Review · Reviewer_daQc · 2025-11-01

**Soundness:** 3
**Presentation:** 3
**Contribution:** 3
**Rating:** 6
**Confidence:** 3

**Summary:**

This paper presents ReDDiT, a novel discrete visual generative model that improves upon traditional discrete diffusion models by introducing rehashing noise. This innovation addresses limitations in absorbing states and sampling diversity, expanding latent variable paths to enhance both diversity and image quality. The model outperforms existing approaches, including MaskGIT, with significant gains in gFID and Inception Score.

**Strengths:**

1. The introduction of rehashing noise provides a novel way to enrich latent variable traversal, offering improved diversity and higher quality generation in discrete diffusion models.

2. ReDDiT outperforms the baseline models (MaskGIT and DDM) on critical metrics like gFID and IS, with competitive efficiency when compared to continuous models.

3. The model works effectively with large vocabulary codebooks (up to 16,384 entries), demonstrating its robustness even when scaled.

**Weaknesses:**

1. Although ReDDiT reports strong numbers on ImageNet‑1K, its effectiveness on more complex or diverse datasets is not discussed. Moreover, for a generation paper, the visualization evidence is quite limited, making it difficult to fully assess the qualitative improvements or appreciate the contribution beyond the single benchmark.

2. The paper does not discuss potential limitations or failure cases of rehashing noise, especially under large‑vocabulary tokenizers or more difficult semantic distributions. Without such analysis, the robustness and stability of the approach remain unclear.

3. How does the rehashing noise behave when scaling to far larger category spaces or higher‑complexity images? Is there a regime where the increased latent path diversity starts to hurt performance (e.g., over‑randomization), and if so, how is this controlled or prevented?

**Questions:**

See the weakness part.

---

> ### Author Response · Authors · 2025-11-24
>
> Thank you for the insightful questions regarding evaluation beyond ImageNet-1K and the behavior of rehashed noise under more challenging semantic conditions. We carefully respond to your questions point-to-point below.
>
> ___
>
> **For Visualization Comment**
>
> Thank you for pointing this out. As suggested, **we have added more qualitative examples to the main paper and appendix** to help reviewers more fully assess the visual improvements brought by ReDDiT.
>
> ____
>
> **1. On evaluating ReDDiT on more complex or diverse datasets**
>
> We appreciate the reviewer’s suggestion. ImageNet-1K remains the most widely adopted benchmark for class-conditional image generation (e.g., ADM, DiT, U-ViT). Our experiments already included L, XL, and XL-F8 (1024-token) models, each trained for over 500k steps, which require more than a week on multiple A100 GPUs. Given the substantial computational cost, scaling to datasets significantly larger than ImageNet-1K is challenging to us with limited GPUs during the rebuttal period. We will start the experiment and expect to include the results to the camera-ready version.
>
> That said, the architectural change introduced by ReDDiT is dataset-agnostic and does not rely on ImageNet-specific priors. To further demonstrate its robustness, we designed additional stress-tests that better simulate complex semantic conditions (see **Point 2**).
>
> ___
>
> **2. On the effectiveness of rehashing noise under more difficult semantic distributions**
>
> Thank you for this constructive comment. Although we have difficulty to train on significantly larger datasets during rebuttal, we perform controlled experiments to emulate scenarios where semantic conditioning becomes **substantially harder**.
>
> (a)  Reduced-capacity semantic modulation (*ReDDiT-shrink*)
>
> ReDDiT uses a standard DiT architecture where semantic conditioning mainly flows through the AdaLN layers (≈30% of transformer parameters). For a 350M model this accounts for ~100M parameters. To simulate challenging semantic distributions, we reduce the AdaLN dimension from 1024 to 128, compressing the semantic modulation capacity by 8×, while keeping all other layers unchanged.
>
> (b)  Removing AdaLN and using in-context conditioning (*ReDDiT-context*)
>
> We further push difficulty by removing AdaLN entirely and feeding class labels only as a prepended token, following an in-context conditioning strategy known to be considerably harder for semantic alignment. This setting stresses the model's ability to handle semantic control without explicit modulation layers.
>
> | Method| Capacity | #Parameters | gFID $\downarrow$ | IS $\uparrow$ |
> |---|---|---|---|---|
> | ReDDiT-shrink | m=1 |  270M | 3.86 | 260.1 |
> | ReDDiT-shrink | m=128 | 270M | 3.46 | 272.1 |
> | ReDDiT-context | m=1 | 229M | 4.01 | 241.4 |
> | ReDDiT-context | m=128 | 229M | 3.71 | 254.9 |
>
> Across both settings, ReDDiT consistently improves generative quality, demonstrating that rehashed noise remains beneficial even when semantic information is highly compressed or weakly provided. These experiments strongly suggest that the mechanism is compatible with harder distributions regimes.
>
> ___
>
> **3. On whether increased latent-path diversity can harm performance**
>
> This is an excellent point. Generation models typically face a quality–diversity tradeoff. While introducing rehash noise improves FID, we observe a minor decrease in IS (≈5 points), which aligns with expectations: higher path diversity may slightly relax mode sharpness.
>
> We note two important observations:
> 1. The effect is easily controlled.
> Adjusting the rehashing strength or disabling rehashing in the later sampling steps mitigates the IS drop. We intentionally avoid detailed hyperparameter tuning to keep the paper concise, but the simplest remedy is to avoid excessively large noise capacity, e.g., setting m=128 as a sweet point.
>
> 2. Rehashing acts as both sampling-time diversification and training-time regularization.
> In the shrink and context experiments above, a moderate noise capacity consistently improves performance, suggesting that rehashed noise helps optimization and reduces over-fitting to specific latent trajectories—a desirable property for scaling to larger models or more diverse datasets.
> ___
> We hope the above clarification addresses your concerns and is helpful for your final assessment. Thank you again for the thoughtful feedback.

---

### Meta-Review · Area_Chair_8743 · 2026-01-10

**Summary:**

This paper introduces ReDDiT, a simple and well-motivated improvement to discrete diffusion that rethinks noise design via rehashing. Reviewers generally agreed the idea is novel, theoretically sound, and leads to very strong gains for discrete diffusion, closing much of the gap to continuous models while keeping the efficiency benefits. Most feedback was positive, with a few questions around clarity and robustness.

**Reviewer Concerns:**

The rebuttal addressed the main concerns well. The authors clarified the rehashing noise and sampler, fixed notation issues, added more qualitative results, and included stress tests under harder semantic settings. Questions around stochasticity control, tokenizer-dependent tuning, and sampling-step behavior were discussed and backed up with additional experiments. Some open questions remain around scaling to much larger datasets and fully matching the very best continuous models, but these are not blockers.

**Reviewer Scores:**

The rebuttal went pretty well. i didn't expect much change.

---

### Decision · Program_Chairs · 2026-01-26

Accept (Poster)